# PEAR: Primitive enabled Adaptive Relabeling for boosting Hierarchical Reinforcement Learning

**Utsav Singh**
CSE Deptt.
IIT Kanpur, India
utsavz@iitk.ac.in

**Vinay P Namboodiri**
CS Deptt.
University of Bath, Bath, UK
vpn22@bath.ac.uk

## ABSTRACT

Hierarchical reinforcement learning (HRL) has the potential to solve complex long horizon tasks using temporal abstraction and increased exploration. However, hierarchical agents are difficult to train due to inherent non-stationarity. We present primitive enabled adaptive relabeling (PEAR), a two-phase approach where we first perform adaptive relabeling on a few expert demonstrations to generate efficient subgoal supervision, and then jointly optimize HRL agents by employing reinforcement learning (RL) and imitation learning (IL). We perform theoretical analysis to bound the sub-optimality of our approach and derive a joint optimization framework using RL and IL. Since PEAR utilizes only a few expert demonstrations and considers minimal limiting assumptions on the task structure, it can be easily integrated with typical off-policy RL algorithms to produce a practical HRL approach. We perform extensive experiments on challenging environments and show that PEAR is able to outperform various hierarchical and non-hierarchical baselines and achieve upto $80\%$ success rates in complex sparse robotic control tasks where other baselines typically fail to show significant progress. We also perform ablations to thoroughly analyse the importance of our various design choices. Finally, we perform real world robotic experiments on complex tasks and demonstrate that PEAR consistently outperforms the baselines.

## 1 INTRODUCTION

Reinforcement learning has been successfully applied to a number of short-horizon robotic manipulation tasks (Rajeswaran et al., 2018; Kalashnikov et al., 2018; Gu et al., 2017; Levine et al., 2016). However, solving long horizon tasks requires long-term planning and is hard (Gupta et al., 2019) due to inherent issues like credit assignment and ineffective exploration. Consequently, such tasks require a large number of environment interactions for learning, especially in sparse reward scenarios (Andrychowicz et al., 2017). Hierarchical reinforcement learning (HRL) (Sutton et al., 1999; Dayan and Hinton, 1993; Vezhnevets et al., 2017; Klissarov et al., 2017; Bacon et al., 2017) is an elegant framework that employs temporal abstraction and promises improved exploration (Nachum et al., 2019). In goal-conditioned feudal architecture (Dayan and Hinton, 1993; Vezhnevets et al., 2017), the higher policy predicts subgoals for the lower primitive, which in turn tries to achieve these subgoals by executing primitive actions directly on the environment. Unfortunately, HRL suffers from non-stationarity (Nachum et al., 2018; Levy et al., 2018) in off-policy HRL. Due to continuously changing policies, previously collected off-policy experience is rendered obsolete, leading to unstable higher level state transition and reward functions.

Some hierarchical approaches (Gupta et al., 2019; Fox et al., 2017; Krishnan et al., 2019) segment the expert demonstrations into subgoal transition dataset, and consequently leverage the subgoal dataset to bootstrap learning. Ideally, the segmentation process should produce subgoals that properly balance the task split between hierarchical levels. One possible approach of task segmentation is to perform fixed window based relabeling (Gupta et al., 2019) on expert demonstrations. Despite being simple, this is effectively a brute force segmentation approach which may generate subgoals that

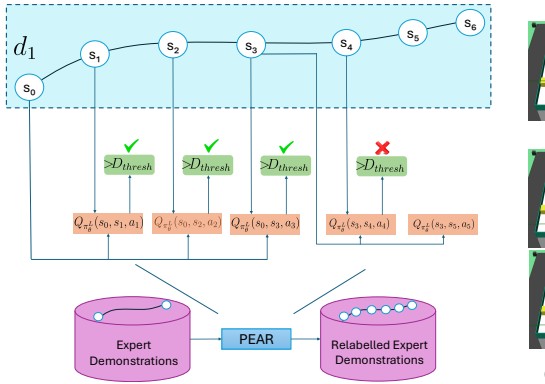

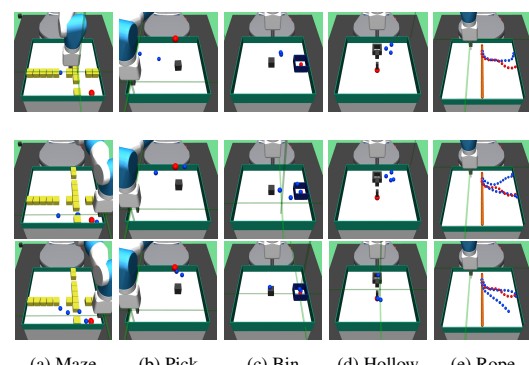

(a) Maze   (b) Pick   (c) Bin   (d) Hollow   (e) Rope

Figure 1: **Adaptive Relabeling Overview**: We segment expert demonstrations by consecutively passing demonstration states as subgoals to the lower primitive, and finding the state where $Q_{\pi_L}(s, s_i, a_i) < Q_{thresh}$ (here $s_i = s_4$). Since $s_3$ was the last reachable subgoal, it is selected as subgoal for initial state $s_0$. The transition is added to $D_g$, and the process continues with $s_3$ as new initial state.

Figure 2: **Subgoal evolution**: With training, as the lower primitive improves, the higher level subgoal predictions (blue spheres) become better and harder, while always being achievable by lower primitive. Row 1 depicts initial training, Row 2 depicts mid-way through training, and Row 3 depicts end of training. This generates a curriculum of achievable subgoals for lower primitive (red spheres: final goal).

are either too easy or too hard according to the current goal achieving ability of the lower primitive, thus leading to degenerate solutions.

The main motivation of this work is to produce a curriculum of feasible subgoals according to the current goal achieving capability of the lower primitive. Concretely, the value function of the lower primitive is used to perform *adaptive relabeling* on expert demonstrations to dynamically generate a curriculum of achievable subgoals for the lower primitive. This subgoal dataset is then used to train an imitation learning based regularizer, which is used to jointly optimize off-policy `RL` objective with `IL` regularization. Hence, our approach ameliorates non-stationarity in `HRL` by using primitive enabled `IL` regularization, while enabling efficient exploration using `RL`. We call our approach: *primitive enabled adaptive relabeling* (`PEAR`) for boosting `HRL`.

The major contributions of this work are: $(i)$ our adaptive relabeling based approach generates efficient higher level subgoal supervision according to the current goal achieving capability of the lower primitive (Figure 2), $(ii)$ we derive sub-optimality bounds to theoretically justify the benefits of periodic re-population using adaptive relabeling (Section 4.3), $(iii)$ we perform extensive experimentation on sparse robotic tasks: maze navigation, pick and place, bin, hollow, rope manipulation and franka kitchen to empirically demonstrate superior performance and sample efficiency of `PEAR` over prior baselines (Section 5 Figure 3), and finally, $(iv)$ we show that `PEAR` demonstrates impressive performance in real world tasks: pick and place, bin and rope manipulation (Figure 6).

## 2 RELATED WORK

Hierarchical reinforcement learning (`HRL`) (Barto and Mahadevan, 2003; Sutton et al., 1999; Parr and Russell, 1998; Dietterich, 2000) promises the advantages of temporal abstraction and increased exploration (Nachum et al., 2019). The options architecture (Sutton et al., 1999; Bacon et al., 2017; Harutyunyan et al., 2018; Harb et al., 2018; Harutyunyan et al., 2019; Klissarov et al., 2017) learns temporally extended macro actions and a termination function to propose an elegant hierarchical framework. However, such approaches may produce degenerate solutions in the absence of proper regularization. Some approaches restrict the problem search space by greedily solving for specific goals (Kaelbling, 1993; Foster and Dayan, 2002), which has also been extended to hierarchical `RL` (Wulfmeier et al., 2019; 2021; Ding et al., 2019). In goal-conditioned feudal learning (Dayan and Hinton, 1993; Vezhnevets et al., 2017), the higher level agent produces subgoals for the lower primitive, which in turn executes atomic actions on the environment. Unfortunately, off-policy `HRL` approaches are cursed by non-stationarity issue. Prior works (Nachum et al., 2018; Levy et al., 2018) deal with the non-stationarity by relabeling previously collected transitions for training goal-conditioned policies. In contrast, our proposed approach deals with non-stationarity by leveraging adaptive relabeling for *periodically* producing achievable subgoals, and subsequently using an im-

itation learning based regularizer. We empirically show in Section 5 that our regularization based approach outperforms relabeling based hierarchical approaches on various long horizon tasks.

Prior methods (Rajeswaran et al., 2018; Nair et al., 2018; Hester et al., 2018) leverage expert demonstrations to improve sample efficiency and accelerate learning, where some methods use imitation learning to bootstrap learning (Shiarlis et al., 2018; Krishnan et al., 2017; 2019; Kipf et al., 2019). Some approaches use fixed relabeling (Gupta et al., 2019) for performing task segmentation. However, such approaches may cause unbalanced task split between hierarchical levels. In contrast, our approach sidesteps this limitation by properly balancing hierarchical levels using adaptive relabeling. Intuitively, we enable balanced task split, thereby avoiding degenerate solutions. Recent approaches restrict subgoal space using adjacency constraints (Zhang et al., 2020), employ graph based approaches for decoupling task horizon (Lee et al., 2023), or incorporate imagined subgoals combined with KL-constrained policy iteration scheme (Chane-Sane et al., 2021). However, such approaches assume additional environment constraints and only work on relatively shorter horizon tasks with limited complexity. (Kreidieh et al., 2020) is an inter-level cooperation based approach for generating achievable subgoals, However, the approach requires extensive exploration for selecting good subgoals, whereas our approach rapidly enables effective subgoal generation using primitive enabled adaptive relabeling. In order to accelerate RL, recent works firstly learn behavior skill priors (Pertsch et al., 2020; Singh et al., 2021) from expert data or pre-train policies over a related task, and then fine-tune using RL. Such approaches largely depend on policies learnt during pre-training, and are hard to train when the source and target task distributions are dissimilar. Other approaches either use bottleneck option discovery (Salter et al., 2022b) or behavior priors (Salter et al., 2022a; Tirumala et al., 2022) to discover and embed behaviors from past experience, or directly hand-design action primitives (Dalal et al., 2021; Nasiriany et al., 2022). While this simplifies the higher level task, explicitly designing action primitives is tedious for hard tasks, and may lead to sub-optimal predictions. Since PEAR learns multi-level policies in parallel, the lower level policies can learn required optimal behavior, thus avoiding the issues with prior approaches.

## 3 BACKGROUND

**Off-policy Reinforcement Learning** We define our goal-conditioned off-policy RL setup as follows: *Universal Markov Decision Process* (UMDP) (Schaul et al., 2015) is a Markov decision process augmented with the goal space $G$, where $M = (S, A, P, R, \gamma, G)$. Here, $S$ is state space, $A$ is action space, $P(s'|s, a)$ is the state transition probability function, $R$ is reward function, and $\gamma$ is discount factor. $\pi(a|s, g)$ represents the goal-conditioned policy which predicts the probability of taking action $a$ when the state is $s$ and goal is $g$. The overall objective is to maximize expected future discounted reward distribution: $J = (1 - \gamma)^{-1} \mathbb{E}_{s \sim d^\pi, a \sim \pi(a|s,g), g \sim G} [r(s_t, a_t, g)]$.

**Hierarchical Reinforcement Learning** In our goal-conditioned HRL setup, the overall policy $\pi$ is divided into multi-level policies. We consider bi-level scheme, where the higher level policy $\pi^H(s_g|s, g)$ predicts subgoals $s_g$ for the lower primitive $\pi^L(a|s, s_g)$. $\pi^H$ generates subgoals $s_g$ after every $c$ timesteps and $\pi^L$ tries to achieve $s_g$ within $c$ timesteps. $\pi^H$ gets sparse extrinsic reward $r_{ex}$ from the environment, whereas $\pi^L$ gets sparse intrinsic reward $r_{in}$ from $\pi^H$. $\pi^L$ gets rewarded with reward $0$ if the agent reaches within $\delta^L$ distance of the predicted subgoal $s_g$, and $-1$ otherwise: $r_{in} = -1(\|s_t - s_g\|_2 > \delta^L)$. Similarly, $\pi^H$ gets extrinsic reward $0$ if the achieved goal is within $\delta^H$ distance of the final goal $g$, and $-1$ otherwise: $r_{ex} = -1(\|s_t - g\|_2 > \delta^H)$. We assume access to a small number of directed expert demonstrations $D = \{e^i\}_{i=1}^N$, where $e^i = (s_0^e, a_0^e, s_1^e, a_1^e \ldots, s_{T-1}^e, a_{T-1}^e)$.

**Limitations of existing approaches to HRL** Off-policy HRL promises the advantages of temporal abstraction and improved exploration (Nachum et al., 2019). Unfortunately, HRL approaches suffer from non-stationarity due to unstable lower primitive. Consequently, HRL approaches fail to perform in complex long-horizon tasks, especially when the rewards are sparse. The primary motivation of this work is to efficiently leverage a few expert demonstrations to bootstrap RL using IL regularization, and thus devise an efficient HRL approach to mitigate non-stationarity.

## 4 METHODOLOGY

In this section, we explain PEAR: **P**rimitive **E**nabled **A**daptive **R**elabeling for boosting HRL, which leverages a few expert demonstrations $D$ to solve long horizon tasks. We propose a two step approach: $(i)$ the current lower primitive $\pi^L$ is used to adaptively relabel expert demonstrations to generate efficient subgoal supervision $D_g$, and $(ii)$ off-policy RL objective is jointly optimized with additional imitation learning based regularization objective using $D_g$. We also perform theoretical analysis to $(i)$ bound the sub-optimality of our approach, and $(ii)$ propose a practical generalized based framework for joint optimization using RL and IL, where typical off-policy RL and IL algorithms can be plugged in to generate various joint optimization based algorithms.

---

**Algorithm 1** Adaptive Relabeling

1: Initialize $D_g = \{\}$
2: // Populating $D_g$
3: **for** each $e = (s_0^e, s_1^e, \ldots, s_{T-1}^e)$ in $\mathcal{D}$ **do**
4:     Initial state index $init \leftarrow 0$
5:     Subgoal transitions $D_g^e = \{\}$
6:     **for** i = 1 **to** $T - 1$ **do**
7:         // Find $Q_{\pi^L}$ values for demo subgoals
8:         Compute $Q_{\pi^L}(s_{init}^e, s_i^e, a_i)$
9:           where $a_i = \pi^L(s_{i-1}^e, s_i^e)$
10:         // Find first subgoal s.t. $Q_{\pi^L} < Q_{th}$
11:         **if** $Q_{\pi^L}(s_{init}^e, s_i^e, a_i) < Q_{th}$ **then**
12:             **for** j = $init$ **to** $i - 1$ **do**
13:                 **for** k = $(init + 1)$ **to** $i - 1$ **do**
14:                     // Add the transition to $D_g^e$
15:                     Add $(s_j, s_{i-1}, s_k)$ to $D_g^e$
16:             Initial state index $init \leftarrow (i - 1)$
17:     // Add selected transitions to $D_g$
18:     $D_g \leftarrow D_g \cup D_g^e$

---

**Algorithm 2** PEAR

1: Initialize $D_g = \{\}$
2: **for** $i = 1 \ldots N$ **do**
3:     **if** $i\%p == 0$ **then**
4:         Clear $D_g$
5:         Populate $D_g$ via adaptive relabeling
6:     Collect experience using $\pi^H$ and $\pi^L$
7:     Update lower primitive via SAC and IL
8:         regularization with $D_g^L$ (Eq 6 or Eq 8)
9:     Sample transitions from $D_g$
10:     Update higher policy via SAC and IL
11:         regularization using $D_g$ (Eq 5 or Eq 7)

---

### 4.1 PRIMITIVE ENABLED ADAPTIVE RELABELING

PEAR performs adaptive relabeling on expert demonstration trajectories $D$ to generate efficient higher level subgoal transition datatset $D_g$, by employing the *current* lower primitive's action value function $Q_{\pi^L}(s, s_i^e, a_i)$. In a typical goal-conditioned RL setting, $Q_{\pi^L}(s, s_i^e, a_i)$ describes the expected cumulative reward where the start state and subgoal are $s$ and $s_i^e$, and the lower primitive takes action $a_i$ while following policy $\pi^L$. While parsing $D$, we consecutively pass the expert demonstrations states $s_i^e$ as subgoals, and $Q_{\pi^L}(s, s_i^e, a_i)$ computes the expected cumulative reward when the start state is $s$, subgoal is $s_i^e$ and the next primitive action is $a_i$. Intuitively, a high value of $Q_{\pi^L}(s, s_i^e, a_i)$ implies that the current lower primitive considers $s_i^e$ to be a good (highly rewarding and achievable) subgoal from current state $s$, since it expects to achieve a high intrinsic reward for this subgoal from the higher policy. Hence, $Q_{\pi^L}(s, s_i^e, a_i)$ considers goal achieving capability of current lower primitive for populating $D_g$. We depict a single pass of adaptive relabeling in Figure 1 and explain the procedure in detail below.

**Adaptive Relabeling** Consider the demonstration dataset $D = \{e^j\}_{i=1}^N$, where each trajectory $e^j = (s_0^e, s_1^e, \ldots, s_{T-1}^e)$. Let the initial state be $s_0^e$. In the adaptive relabeling procedure, we incrementally provide demonstration states $s_i^e$ for $i = 1$ to $T - 1$ as subgoals to lower primitive's action value function $Q_{\pi^L}(s_0^e, s_i^e, a_i)$, where $a_i = \pi^L(s = s_{i-1}^e, g = s_i^e)$. At every step, we compare $Q_{\pi^L}(s_0^e, s_i^e, a_i)$ to a threshold $Q_{thresh}$ ($Q_{thresh} = 0$ works consistently for all experiments). If $Q_{\pi^L}(s_0^e, s_i^e, a_i) >= Q_{thresh}$, we move on to next expert demonstration state $s_{i+1}^e$. Otherwise, we consider $s_{i-1}^e$ to be a good subgoal (since it was the last subgoal with $Q_{\pi^L}(s_0^e, s_{i-1}^e, a_i) >= Q_{thresh}$), and use it to compute subgoal transitions for populating $D_g$. Sub-

sequently, we repeat the same procedure with $s_{i-1}^e$ as the new initial state, until the episode terminates. This is depicted in Figure 1 and Algorithm 1.

**Periodic re-population of higher level subgoal dataset** HRL approaches suffer from non-stationarity due to unstable higher level station transition and reward functions. In off-policy HRL, this occurs since previously collected experience is rendered obsolete due to continuously changing lower primitive. We propose to mitigate this non-stationarity by periodically re-populating subgoal transition dataset $D_g$ after every $p$ timesteps according to the goal achieving capability of the *current* lower primitive. Since the lower primitive continuously improves with training and gets better at achieving harder subgoals, $Q_{\pi^L}$ always picks reachable subgoals of appropriate difficulty, according to the current lower primitive. This generates a natural curriculum of achievable subgoals for the lower primitive. Intuitively, $D_g$ always contains achievable subgoals for the current lower primitive, which mitigates non-stationarity in HRL. The pseudocode for PEAR is given in Algorithm 2. Figure 2 shows the qualitative evolution of subgoals with training in our experiments.

**Dealing with out-of-distribution states** Our adaptive relabeling procedure uses $Q_{\pi^L}(s_0^e, s_i^e, a_i)$ to select efficient subgoals when the expert state $s_i^e$ is within the training distribution of states used to train the lower primitive. However, if the expert states are outside the training distribution, $Q_{\pi_L}$ might erroneously over-estimate the values on such states, which might result in poor subgoal selection. In order to address this over-estimation issue, we employ an additional margin classification objective(Piot et al., 2014), where along with the standard $Q_{SAC}$ objective, we also use an additional margin classification objective to yield the following optimization objective $\bar{Q}_{\pi^L} = Q_{SAC} +$

$$\arg\min_{Q_{\pi_L}} \max_{\pi^L}(\mathbb{E}_{(s_0^e, \cdot, \cdot) \sim D_g, s_i^e \sim \pi^H, a_i \sim \pi^L}[Q_{\pi^L}(s_0^e, s_i^e, a_i)] - \mathbb{E}_{(s_0^e, s_i^e, \cdot) \sim D_g, a_i \sim \pi^L}[Q_{\pi^L}(s_0^e, s_i^e, a_i)])$$

This surrogate objective prevents over-estimation of $\bar{Q}_{\pi^L}$ by penalizing states that are out of the expert state distribution. We found this objective to improve performance and stabilize learning. In the next section, we explain how we use adaptive relabeling to yield our joint optimization objective.

## 4.2 JOINT OPTIMIZATION

Here, we explain our joint optimization objective that comprises of off-policy RL objective with IL based regularization, using $D_g$ generated using primitive enabled adaptive relabeling. We consider both behavior cloning (BC) and inverse reinforcement learning (IRL) regularization. Henceforth, PEAR-IRL will represent PEAR with IRL regularization and PEAR-BC will represent PEAR with BC regularization. We first explain BC regularization objective, and then explain IRL regularization objectives for both hierarchical levels.

For the BC objective, let $(s^e, s_g^e, s_{next}^e) \sim D_g$ represent a higher level subgoal transition from $D_g$ where $s^e$ is current state, $s_{next}^e$ is next state, $g^e$ is final goal and $s_g^e$ is subgoal supervision. Let $s_g$ be the subgoal predicted by the high level policy $\pi_{\theta_H}^H(\cdot|s^e, g^e)$ with parameters $\theta_H$. The BC regularization objective for higher level is as follows:

$$\min_{\theta_H} J_{BC}^H(\theta_H) = \min_{\theta_H} \mathbb{E}_{(s^e, s_g^e, s_{next}^e) \sim D_g, s_g \sim \pi_{\theta_H}^H(\cdot|s^e, g^e)} ||s_g^e - s_g||^2 \tag{1}$$

Similarly, let $(s^f, a^f, s_{next}^f) \sim D_g^L$ represent lower level expert transition where $s^f$ is current state, $s_{next}^f$ is next state, $g^f$ is goal and $a$ is the primitive action predicted by $\pi_{\theta_L}^L(\cdot|s^f, s_g^e)$ with parameters $\theta_L$. The lower level BC regularization objective is as follows:

$$\min_{\theta_L} J_{BC}^L(\theta_L) = \min_{\theta_L} \mathbb{E}_{(s^f, a^f, s_{next}^f) \sim D_g^L, a \sim \pi_{\theta_L}^L(\cdot|s^f, s_g^e)} ||a^f - a||^2 \tag{2}$$

We now consider the IRL objective, which is implemented as a GAIL (Ho and Ermon, 2016) objective implemented using LSGAN (Mao et al., 2016). Let $\mathbb{D}_{\epsilon}^H$ be the higher level discriminator with parameters $\epsilon_H$. Let $J_D^H$ represent higher level IRL objective, which depends on parameters $(\theta_H, \epsilon_H)$. The higher level IRL regularization objective is as follows:

$$\max_{\theta_H} \min_{\epsilon_H} J_D^H(\theta_H, \epsilon_H) = \max_{\theta_H} \min_{\epsilon_H} \frac{1}{2} \mathbb{E}_{(s^e, \cdot, \cdot) \sim D_g, s_g \sim \pi_{\theta_H}(\cdot|s^e, g^e)}[\mathbb{D}_{\epsilon_H}^H(\pi_{\theta_H}^H(\cdot|s^e, g^e)) - 0]^2$$
$$+ \frac{1}{2} \mathbb{E}_{(s^e, s_g^e, \cdot) \sim D_g}[\mathbb{D}_{\epsilon_H}^H(s_g^e) - 1]^2 \tag{3}$$

Similarly, for lower level primitive, let $\mathbb{D}_{\epsilon_L}^L$ be the lower level discriminator with parameters $\epsilon_L$. Let $J_D^L$ represent lower level IRL objective, which depends on parameters $(\theta_L, \epsilon_L)$. The lower level IRL regularization objective is as follows:

$$\max_{\theta_L} \min_{\epsilon_L} J_D^L(\theta_L, \epsilon_L) = \max_{\theta_L} \min_{\epsilon_L} \frac{1}{2}\mathbb{E}_{(s^f, \cdot, \cdot) \sim D_g^L, a \sim \pi_{\theta_L}^L(\cdot|s^f, s_g^e)}[\mathbb{D}_{\epsilon_L}^L(\pi_{\theta_L}^L(\cdot|s^f, s_g^e)) - 0]^2$$
$$+ \frac{1}{2}\mathbb{E}_{(s^f, a^f, \cdot) \sim D_g^L}[\mathbb{D}_{\epsilon_L}^L(a^f) - 1]^2 \quad (4)$$

Finally, we describe our joint optimization objective for hierarchical policies. Let the off-policy RL objective be $J_{\theta_H}^H$ and $J_{\theta_L}^L$ for higher and lower policies. The joint optimization objectives using BC regularization for higher and lower policies are provided in Equations 5 and 6 respectively.

$$\max_{\theta_H}(J_{\theta_H}^H - \psi * J_{BC}^H(\theta_H)) \quad (5)$$

$$\max_{\theta_L}(J_{\theta_L}^L - \psi * J_{BC}^L(\theta_L)) \quad (6)$$

The joint optimization objectives using IRL regularization for higher and lower policies are provided in Equations 7 and 8 respectively.

$$\min_{\epsilon_H} \max_{\theta_H}(J_{\theta_H}^H + \psi * J_D^H(\theta_H, \epsilon_H)) \quad (7)$$

$$\min_{\epsilon_L} \max_{\theta_L}(J_{\theta_L}^L + \psi * J_D^L(\theta_L, \epsilon_L)) \quad (8)$$

Here, $\psi$ is regularization weight hyper-parameter. We describe ablations to choose $\psi$ in Section 5.

## 4.3 SUB-OPTIMALITY ANALYSIS

In this section, we perform theoretical analysis to $(i)$ derive sub-optimality bounds for our proposed joint optimization objective and show how our periodic re-population based approach affects performance, and $(ii)$ propose a generalized framework for joint optimization using RL and IL. Let $\pi^*$ and $\pi^{**}$ be unknown higher level and lower level optimal policies. Let $\pi_{\theta_H}^H$ be our high level policy and $\pi_{\theta_L}^L$ be our lower primitive policy, where $\theta_H$ and $\theta_L$ are trainable parameters. $D_{TV}(\pi_1, \pi_2)$ denotes total variation divergence between probability distributions $\pi_1$ and $\pi_2$. Let $\kappa$ be an unknown distribution over states and actions, $G$ be goal space, $s$ be current state, and $g$ be the final episodic goal. We use $\kappa$ in the importance sampling ratio later to avoid sampling from the unknown optimal policy (Appendix A.1). The higher level policy predicts subgoals $s_g$ for the lower primitive which is executed for $c$ timesteps to yield sub-trajectories $\tau$. Let $\Pi_D^H$ and $\Pi_D^L$ be some unknown higher and lower level probability distributions over policies from which we can sample policies $\pi_D^H$ and $\pi_D^L$. Let us assume that policies $\pi_D^H$ and $\pi_D^L$ represent the policies from higher and lower level datasets $D_H$ and $D_L$ respectively. Although $D_H$ and $D_L$ may represent any datasets, in our discussion, we use them to represent higher and lower level expert demonstration datasets. Firstly, we introduce the $\phi_D$-common definition (Ajay et al., 2020) in goal-conditioned policies:

**Definition 1.** $\pi^*$ is $\phi_D$-common in $\Pi_D^H$, if $\mathbb{E}_{s \sim \kappa, \pi_D^H \sim \Pi_D^H, g \sim G}[D_{TV}(\pi^*(\tau|s, g)||\pi_D^H(\tau|s, g))] \leq \phi_D$

Now, we define the suboptimality of policy $\pi$ with respect to optimal policy $\pi^*$ as:

$$Subopt(\theta) = |J(\pi^*) - J(\pi)| \quad (9)$$

**Theorem 1.** *Assuming optimal policy $\pi^*$ is $\phi_D$ common in $\Pi_D^H$, the suboptimality of higher policy $\pi_{\theta_H}^H$, over $c$ length sub-trajectories $\tau$ sampled from $d_c^{\pi^*}$ can be bounded as:*

$$|J(\pi^*) - J(\pi_{\theta_H}^H)| \leq \underbrace{\lambda_H * \phi_D}_{\text{first term}} + \underbrace{\lambda_H * \mathbb{E}_{s \sim \kappa, \pi_D^H \sim \Pi_D^H, g \sim G}[D_{TV}(\pi_D^H(\tau|s, g)||\pi_{\theta_H}^H(\tau|s, g))]}_{\text{second term}} \quad (10)$$

*where $\lambda_H = \frac{2}{(1-\gamma)(1-\gamma^c)}R_{max}\|\frac{d_c^{\pi^*}}{\kappa}\|_\infty$*

*Similarly, the suboptimality of lower primitive $\pi_{\theta_L}^L$ can be bounded as:*

$$|J(\pi^{**}) - J(\pi_{\theta_L}^L)| \leq \lambda_L * \phi_D + \lambda_L * \mathbb{E}_{s \sim \kappa, \pi_D^L \sim \Pi_D^L, s_g \sim \pi_{\theta_H}^H}[D_{TV}(\pi_D^L(\tau|s, s_g)||\pi_{\theta_L}^L(\tau|s, s_g))] \tag{11}$$

*where $\lambda_L = \frac{2}{(1-\gamma)^2} R_{max} \|\frac{d_c^{\pi^{**}}}{\kappa}\|_\infty$*

The proofs for Equations 10 and 11 are provided in Appendix A.1. We next discuss the effect of training on the two terms in RHS of Equation 10, which bound the suboptimality of $\pi_{\theta_H}^H$.

**Effect of adaptive relabeling on sub-optimality bounds** We firstly focus on the first term which is dependent on $\phi_D$. Since we represent the generated subgoal dataset as $D_g$, we replace $\phi_D$ with $\phi_{D_g}$. In Theorem 1, we assume the optimal policy $\pi^*$ to be $\phi_{D_g}$ common in $\Pi_D^H$. Since $\phi_{D_g}$ denotes the upper bound of the expected TV divergence between $\pi^*$ and $\pi_D^H$, $\phi_{D_g}$ provides a quality measure of the subgoal dataset $D_g$ populated using adaptive relabeling. Intuitively, a lower value of $\phi_{D_g}$ implies that the optimal policy $\pi^*$ is closely represented by $D_g$, or alternatively, the samples from $D_g$ are near optimal. As the lower primitive improves with training and is able to achieve harder subgoals, and since $D_g$ is re-populated using the improved lower primitive after every $p$ timesteps, $\pi_{D_g}$ continually gets closer to $\pi^*$, which results in reduced value of $\phi_D$. This implies that due to decreasing first term, the suboptimality bound in Equation 10 gets tighter, and consequently $J(\pi_{\theta_H}^H)$ gets closer to optimal $J(\pi^*)$ objective. Hence, our periodic re-population based approach generates a natural curriculum of achievable subgoals for the lower primitive, which continuously improves the performance by tightening the upper bound.

**Effect of IL regularization on sub-optimality bounds** Now, we focus on the second term in Equation 10, which is TV divergence between $\pi_D^H(\tau|s, g)$ and $\pi_{\theta_H}^H(\tau|s, g)$ with expectation over $s \sim \kappa, \pi_D^H \sim \Pi_D^H, g \sim G$. As before, $D$ is replaced by dataset $D_g$. This term can be viewed as imitation learning (IL) objective between expert demonstration policy $\pi_{D_g}^H$ and current policy $\pi_{\theta_H}^H$, where TV divergence is the distance measure. Due to this IL regularization objective, as policy $\pi_{\theta_H}^H$ gets closer to expert distribution policy $\pi_{D_g}^H$ with training, the LHS sub-optimality bounds get tighter. Thus, our proposed periodic IL regularization using $D_g$ tightens the sub-optimality bounds in Equation 10 with training, thereby improving performance.

**Generalized framework** We now derive our generalized framework for the joint optimization objective, where we can plug in off-the-shelf RL and IL methods to yield a generally applicable practical HRL algorithm. Considering sub-optimality is positive (Equation 9), we can use Equation 10 to derive the following objective:

$$J(\pi^*) \geq \underbrace{J(\pi_{\theta_H}^H)}_{\text{RL term}} - \underbrace{\lambda_H * \phi_D}_{\text{const. wrt } D_g} - \underbrace{\lambda_H * \mathbb{E}_{s \sim \kappa, \pi_D^H \sim \Pi_D^H, g \sim G}[d(\pi_D^H(\tau|s, g)||\pi_{\theta_H}^H(\tau|s, g))]}_{\text{IL regularization term}} \tag{12}$$

*where (considering $\pi_D^H(\tau|s, g)$ as $\pi_A$, and $\pi_{\theta_H}^H(\tau|s, g)$ as $\pi_B$), $d(\pi_A||\pi_B) = D_{TV}(\pi_A||\pi_B)$.*

Notably, the second term $\lambda_H * \phi_D$ in RHS of Equation 12 is constant for a given dataset $D_g$. Equation 12 can be perceived as a minorize maximize algorithm which intuitively means: the overall objective can be optimized by $(i)$ maximizing the objective $J(\pi_{\theta_H}^H)$ via RL, and $(ii)$ minimizing the distance measure $d$ between $\pi_D^H$ and $\pi_{\theta_H}^H$ (IL regularization). This formulation serves as a framework where we can plug in RL algorithm of choice for off-policy RL objective $J(\pi_{\theta_H}^H)$, and distance function $d$ of choice for IL regularization, to yield various joint optimization objectives.

In our setup, we plug in entropy regularized Soft Actor Critic (Haarnoja et al., 2018a) to maximize $J(\pi_{\theta_H}^H)$. Notably, different parameterizations of $d$ yield different imitation learning regularizers. When $d$ is formulated as Kullback–Leibler divergence, the IL regularizer takes the form of behavior cloning (BC) objective (Nair et al., 2018) (which results in PEAR-BC), and when $d$ is formulated as Jensen-Shannon divergence, the imitation learning objective takes the form of inverse reinforcement learning (IRL) objective (which results in PEAR-IRL). We consider both these objectives in Section 5 and provide empirical performance results.

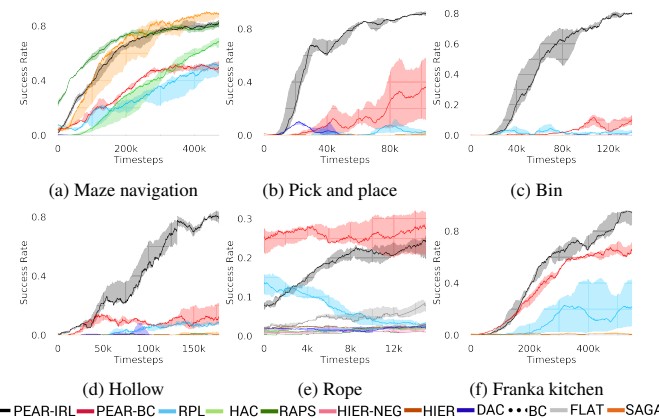

Figure 3: **Success rate comparison** This figure compares the success rate performances on six sparse maze navigation and manipulation tasks. The solid line and shaded region represent the mean and range of success rates across 5 seeds. As seen, PEAR shows impressive performance and significantly outperforms the baselines.

## 5 EXPERIMENTS

In this section, we empirically answer the following questions: $(i)$ does adaptive relabeling approach outperform fixed relabeling based approaches, $(ii)$ is PEAR able to mitigate non-stationarity, $(iii)$ does IL regularization boost performance in solving complex long horizon tasks, and $(iv)$ What is the contribution of each of our design choices? We accordingly perform experiments on six Mujoco (Todorov et al., 2012) environments: $(i)$ maze navigation, $(ii)$ pick and place, $(iii)$ bin, $(iv)$ hollow, $(v)$ rope manipulation, and $(vi)$ franka kitchen. Please refer to the supplementary for a video depicting qualitative results, and the implementation code.

**Environment and Implementation Details:** We provide extensive environment and implementation details, including number and procedure of collecting demonstrations in Appendix A.3. Since the environments are sparsely rewarded, they are complex tasks where the agent must explore the environment extensively before receiving any rewards. Unless otherwise stated, we keep the training conditions consistent across all baselines to ascertain fair comparisons, and empirically tune the hyper-parameter values of our method and all other baselines.

### 5.1 EVALUATION AND RESULTS

In Figure 3, we depict the success rate performance of PEAR and compare it with other baselines averaged over 5 seeds. The primary goal of these comparisons is to verify that the proposed approach indeed mitigates non-stationarity and demonstrates improved performance and training stability.

**Comparing with fixed window based approach**

**RPL:** In order to demonstrate the efficacy of adaptive relabeling, we compare our method with Relay Policy Learning (RPL) baseline. RPL (Gupta et al., 2019) uses supervised pre-training followed by relay fine tuning. In order to ascertain fair comparisons, we use an ablation of RPL which does not use supervised pre-training. PEAR outperforms this baseline, which demonstrates that adaptive relabeling outperforms fixed window based relabeling and is crucial for mitigating non-stationarity. Since PEAR and RPL both employ jointly optimizing RL and IL based learning and only differ in adaptive relabeling, it is evident that adaptive relabeling is crucial for generating feasible subgoals.

**Comparing with hierarchical baselines**

**RAPS:** RAPS (Dalal et al., 2021) uses hand designed action primitives at the lower level, where the goal of the upper level is to pick the optimal sequence of action primitives. The performance of such approaches significantly depends on the quality of action primitives, which require substantial effort to hand-design. We found that except maze navigation task, RAPS is unable to perform well on other tasks, which we believe is because selecting appropriate primitive sequences is hard on other harder tasks. Notably, hand designing action primitives is exceptionally complex in environments like rope manipulation. Hence, we do not evaluate RAPS in rope environment.

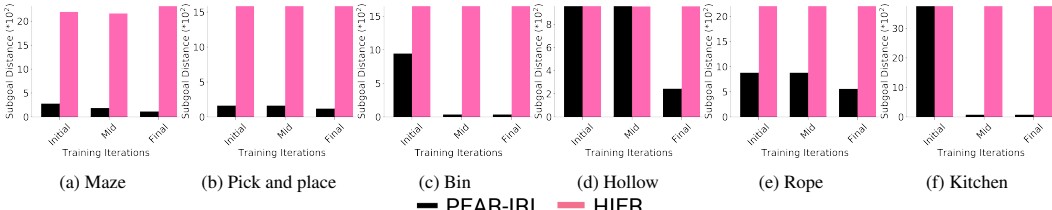

Figure 4: **Non-stationarity metric comparison** This figure compares the average distance metric between the subgoals predicted by the higher level policy and the subgoals achieved by the lower level policy during training. As seen, PEAR consistently produces efficient subgoals leading to low distances between the predicted and achieved subgoals throughout the training process. This mitigates non-stationarity in HRL.

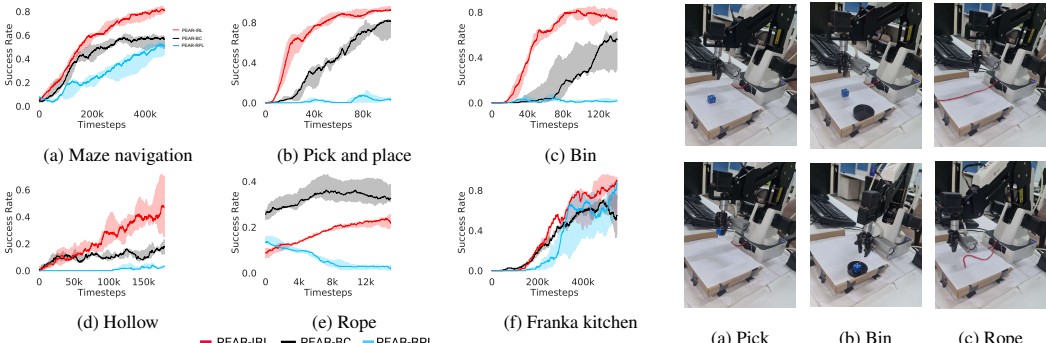

Figure 5: The success rate plots show success rate performance comparison between PEAR-IRL (red), PEAR-BC (black) and PEAR-RPL (blue) ablation. PEAR-IRL and PEAR-BC clearly outperform PEAR-RPL in almost all the tasks.

Figure 6: **Real world experiments** in pick and place, bin and rope environments. Row 1 depicts initial and Row 2 depicts goal configuration.

**HAC:** Hierarchical actor critic (HAC) (Levy et al., 2018) deals with non-stationarity by relabeling transitions while assuming an optimal lower primitive. Although HAC shows good performance in maze navigation task, PEAR consistently outperforms HAC on all other tasks.

**HIER-NEG and HIER:** We also compare PEAR with two hierarchical baselines: HIER and HIER-NEG, which are hierarchical baselines that do not leverage expert demonstrations. HIER-NEG is a hierarchical baseline where the upper level is negatively rewarded if the lower primitive fails to achieve the subgoal. Since HIER, HIER-NEG and PEAR all are hierarchical approaches, we use these baseline comparisons to motivate that the performance improvement is not just due to use of hierarchical abstraction, but instead due to adaptive relabeling and primitive-enabled regularization. This is clearly evidenced by superior performance of PEAR.

### Comparing with non-hierarchical baselines

Additionally, we consider single-level Discriminator Actor Critic (**DAC**) (Kostrikov et al., 2019) that leverages expert demos, single-level SAC (**FLAT**) baseline, and Behavior Cloning (**BC**) baselines. However, they fail to perform well in any of the tasks.

## 5.2 ABLATIVE ANALYSIS

Here, we perform ablation analysis to elucidate the significance of our design choices. We choose the hyper-parameter values after extensive experiments, and keep them consistent across all baselines.

**Dealing with non-stationarity and infeasible subgoal generation in HRL:** We assess whether PEAR mitigates non-stationarity in HRL by comparing it with the vanilla HIER baseline, as shown in Figure 4. We compute the average distance between subgoals predicted by the higher-level policy and those achieved by the lower-level primitive throughout training. A lower average distance suggests that PEAR generates subgoals achievable by the lower primitive, inducing lower primitive behavior to be optimal. Our findings reveal that PEAR consistently maintains low average distances, validating its effectiveness in reducing non-stationarity. Additionally, as seen in Figure 4, post-training results show that PEAR achieves significantly lower distance values than the HIER baseline, highlighting its ability to generate feasible subgoals through primitive regularization.

Figure 7: This figure illustrates the importance of margin surrogate objective by comparing PEAR-IRL and PEAR-BC (with margin objective) with PEAR-IRL-No-Margin and PEAR-BC-No-Margin (without margin objective). PEAR-IRL and PEAR-BC outperform their non margin objective counterparts in almost all tasks.

**Additional Ablations:** We verify the importance of adaptive relabeling by replacing it in `PEAR-IRL` by fixed window relabeling (as in `RPL` (Gupta et al., 2019)). This ablation (`PEAR-IRL`) consistently outperforms `PEAR-RPL` on all tasks (Figure 5), which shows the benefit of adaptive relabeling. Further, we compare `PEAR-IRL` and `PEAR-BC` (with margin classification objectives), with `PEAR-IRL-No-Margin` and `PEAR-BC-No-Margin` (without margin objectives) in Figure 7. `PEAR-IRL` and `PEAR-BC` outperform their `No-Margin` counterparts, which shows that this objective efficiently deals with the issue of out-of-distribution states, and induces training stability.

Further, we analyse how varying $Q_{thresh}$ affects performance in Appendix A.4 Figure 8. We next study the impact of varying $p$. Intuitively, if $p$ is too large, it impedes generation of a good curriculum of subgoals (Appendix A.4 Figure 9). Also, a low value of $p$ may lead to frequent subgoal dataset re-population and may impede stable learning. We also choose optimal window size $k$ for `RPL` experiments, as shown in Appendix A.4 Figure 10. We also evaluate learning rate $\psi$ in Appendix A.4 Figure 11. If $\psi$ is too small, `PEAR` is unable to utilize `IL` regularization, whereas conversely if $\psi$ is too large, the learned policy might overfit. We also deduce the optimal number of expert demonstrations required in Appendix A.4 Figure 12. Next, we compare the performance of PEAR-IRL with HER-BC, which is a single-level implementation of HER with expert demonstrations. As seen in Appendix A.4 Figure 13, PEAR significantly outperforms this baseline, which demonstrates the advantage of our hierarchical formulation. We also provide qualitative visualizations in Appendix A.5.

**Real world experiments:** We perform experiments on real world robotic pick and place, bin and rope environments (Fig 6). We use Realsense D435 depth camera to extract the robotic arm position, block, bin, and rope cylinder positions. Computing accurate linear and angular velocities is hard in real tasks, so we assign them small hard-coded values, which shows good performance. We performed 5 sets of experiments with 10 trial each, and report the average success rates. `PEAR-IRL` achieves accuracy of 0.8, 0.6, and 0.3, whereas `PEAR-BC` achieves accuracy of 0.8, 0, 0.3 on pick and place, bin and rope environments. We also evaluate the performance of next best performing `RPL` baseline, but it fails to achieve success in any of the tasks.

## 6 DISCUSSION

**Limitations** In this work, we assume availability of directed expert demonstrations, which we plan to deal with in future. Additionally, $D_g$ is periodically re-populated, which is an additional overhead and might be a bottleneck in tasks where relabeling cost is high. Notably, we side-step this limitation by passing the whole expert trajectory as a mini-batch for a single forward pass through lower primitive. Nevertheless, we plan to deal with these limitations in future work.

**Conclusion and future work** We propose primitive enabled adaptive relabeling (PEAR), a HRL and IL based approach that performs adaptive relabeling on a few expert demonstrations to solve complex long horizon tasks. We perform comparisons with a various baselines and demonstrate that PEAR shows strong results in simulation and real world robotic tasks. In future work, we plan to address harder sequential decision making tasks, and plan to analyse generalization beyond expert demonstrations. We hope that PEAR encourages future research in the area of adaptive relabeling and primitive informed regularization, and leads to efficient approaches to solve long horizon tasks.

ACKNOWLEDGMENTS

This research work was partially supported by Research-I Foundation of the Department of CSE at IIT Kanpur.

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

CONTENTS

# A APPENDIX

## A.1 SUB-OPTIMALITY ANALYSIS

Here, we present the proofs for Theorem 1 for higher and lower level policies, which provide sub-optimality bounds on the optimization objectives.

### A.1.1 SUB-OPTIMALITY PROOF FOR HIGHER LEVEL POLICY

The sub-optimality of upper policy $\pi_{\theta_H}^H$, over $c$ length sub-trajectories $\tau$ sampled from $d_c^{\pi^*}$ can be bounded as:

$$|J(\pi^*) - J(\pi_{\theta_H}^H)| \leq \lambda_H * \phi_D + \lambda_H * \mathbb{E}_{s\sim\kappa,\pi_D^H\sim\Pi_D^H,g\sim G}[D_{TV}(\pi_D^H(\tau|s,g)||\pi_{\theta_H}^H(\tau|s,g))]] \quad (13)$$

where $\lambda_H = \frac{2}{(1-\gamma)(1-\gamma^c)}R_{max}\|\frac{d_c^{\pi^*}}{\kappa}\|_\infty$

*Proof.* We extend the suboptimality bound from (Ajay et al., 2020) between goal conditioned policies $\pi^*$ and $\pi_{\theta_H}^H$ as follows:

$$|J(\pi^*) - J(\pi_{\theta_H}^H)| \leq \frac{2}{(1-\gamma)(1-\gamma^c)}R_{max}\mathbb{E}_{s\sim d_c^{\pi^*},g\sim G}[D_{TV}(\pi^*(\tau|s,g)||\pi_{\theta_H}^H(\tau|s,g))] \quad (14)$$

By applying triangle inequality:

$$D_{TV}(\pi^*(\tau|s,g)||\pi_{\theta_H}^H(\tau|s,g)) \leq D_{TV}(\pi^*(\tau|s,g)||\pi_D^H(\tau|s,g)) + D_{TV}(\pi_D^H(\tau|s,g)||\pi_{\theta_H}^H(\tau|s,g)) \quad (15)$$

Taking expectation wrt $s\sim\kappa$, $g\sim G$ and $\pi_D^H\sim\Pi_D^H$,

$$\mathbb{E}_{s\sim\kappa,g\sim G}[D_{TV}(\pi^*(\tau|s,g)||\pi_{\theta_H}^H(\tau|s,g))] \leq \mathbb{E}_{s\sim\kappa,\pi_D^H\sim\Pi_D^H,g\sim G}[D_{TV}(\pi^*(\tau|s,g)||\pi_D^H(\tau|s,g))]+$$
$$\mathbb{E}_{s\sim\kappa,\pi_D^H\sim\Pi_D^H,g\sim G}[D_{TV}(\pi_D^H(\tau|s,g)||\pi_{\theta_H}^H(\tau|s,g))] \quad (16)$$

Since $\pi^*$ is $\phi_D$ common in $\Pi_D^H$, we can write 16 as:

$$\mathbb{E}_{s\sim\kappa,g\sim G}[D_{TV}(\pi^*(\tau|s,g)||\pi_{\theta_H}^H(\tau|s,g))] \leq$$
$$\phi_D + \mathbb{E}_{s\sim\kappa,\pi_D^H\sim\Pi_D^H,g\sim G}[D_{TV}(\pi_D^H(\tau|s,g)||\pi_{\theta_H}^H(\tau|s,g))] \quad (17)$$

Substituting the result from align 17 in align 14, we get

$$|J(\pi^*) - J(\pi_{\theta_H}^H)| \leq \lambda_H * \phi_D + \lambda_H * \mathbb{E}_{s\sim\kappa,\pi_D^H\sim\Pi_D^H,g\sim G}[D_{TV}(\pi_D^H(\tau|s,g)||\pi_{\theta_H}^H(\tau|s,g))]] \quad (18)$$

where $\lambda_H = \frac{2}{(1-\gamma)(1-\gamma^c)}R_{max}\|\frac{d_c^{\pi^*}}{\kappa}\|_\infty$ □

### A.1.2 SUB-OPTIMALITY PROOF FOR LOWER LEVEL POLICY

Let the optimal lower level policy be $\pi^{**}$. The suboptimality of lower primitive $\pi_{\theta_L}^L$ can be bounded as follows:

$$|J(\pi^{**}) - J(\pi_{\theta_L}^L)| \leq \lambda_L * \phi_D + \lambda_L * \mathbb{E}_{s\sim\kappa,\pi_D^L\sim\Pi_D^L,s_g\sim\pi_{\theta_H}^H}[D_{TV}(\pi_D^L(\tau|s,s_g)||\pi_{\theta_L}^L(\tau|s,s_g))]] \quad (19)$$

where $\lambda_L = \frac{2}{(1-\gamma)^2}R_{max}\|\frac{d_c^{\pi^{**}}}{\kappa}\|_\infty$

*Proof.* We extend the suboptimality bound from (Ajay et al., 2020) between goal conditioned policies $\pi^{**}$ and $\pi_{\theta_L}^L$ as follows:

$$|J(\pi^{**}) - J(\pi_{\theta_L}^L)| \leq \frac{2}{(1-\gamma)^2}R_{max}\mathbb{E}_{s\sim d_c^{\pi^{**}},s_g\sim\pi_{\theta_H}^H}[D_{TV}(\pi^{**}(\tau|s,s_g)||\pi_{\theta_L}^L(\tau|s,s_g))] \quad (20)$$

By applying triangle inequality:

$$D_{TV}(\pi^{**}(\tau|s,s_g)||\pi_{\theta_L}^L(\tau|s,s_g)) \leq D_{TV}(\pi^{**}(\tau|s,s_g)||\pi_D^L(\tau|s,s_g))+$$
$$D_{TV}(\pi_D^L(\tau|s,s_g)||\pi_{\theta_L}^L(\tau|s,s_g)) \tag{21}$$

Taking expectation wrt $s \sim \kappa$, $s_g \sim \pi_{\theta_H}^H$ and $\pi_D^L \sim \Pi_D^L$,

$$\mathbb{E}_{s\sim\kappa, s_g\sim\pi_{\theta_H}^H}[D_{TV}(\pi^{**}(\tau|s,s_g)||\pi_{\theta_L}^L(\tau|s,s_g))] \leq$$
$$\mathbb{E}_{s\sim\kappa, \pi_D^L\sim\Pi_D^L, s_g\sim\pi_{\theta_H}^H}[D_{TV}(\pi^{**}(\tau|s,s_g)||\pi_D^L(\tau|s,s_g))]+ \tag{22}$$
$$\mathbb{E}_{s\sim\kappa, \pi_D^L\sim\Pi_D^L, s_g\sim\pi_{\theta_H}^H}[D_{TV}(\pi_D^L(\tau|s,s_g)||\pi_{\theta_L}^L(\tau|s,s_g))]$$

Since $\pi^{**}$ is $\phi_D$ common in $\Pi_D^L$, we can write 22 as:

$$\mathbb{E}_{s\sim\kappa, s_g\sim\pi_{\theta_H}^H}[D_{TV}(\pi^{**}(\tau|s,s_g)||\pi_{\theta_L}^L(\tau|s,s_g))] \leq$$
$$\phi_D + \mathbb{E}_{s\sim\kappa, \pi_D^L\sim\Pi_D^L, s_g\sim\pi_{\theta_H}^H}[D_{TV}(\pi_D^L(\tau|s,s_g)||\pi_{\theta_L}^L(\tau|s,s_g))] \tag{23}$$

Substituting the result from align 23 in align 20, we get

$$|J(\pi^{**}) - J(\pi_{\theta_L}^L)| \leq \lambda_L * \phi_D + \lambda_L * \mathbb{E}_{s\sim\kappa, \pi_D^L\sim\Pi_D^L, s_g\sim\pi_{\theta_H}^H}[D_{TV}(\pi_D^L(\tau|s,s_g)||\pi_{\theta_L}^L(\tau|s,s_g))]] \tag{24}$$

where $\lambda_L = \frac{2}{(1-\gamma)^2}R_{max}\|\frac{d_c^{\pi^{**}}}{\kappa}\|_\infty$ ☐

## A.2    GENERATING EXPERT DEMONSTRATIONS

For maze navigation, we use path planning RRT (LaValle, 1998) algorithm to generate expert demonstration trajectories. For pick and place, we hard coded an optimal trajectory generation policy for generating demonstrations, although they can also be generated using Mujoco VR (Todorov et al., 2012). For kitchen task, the expert demonstrations are collected using Puppet Mujoco VR system (Fu et al., 2020). In rope manipulation task, expert demonstrations are generated by repeatedly finding the closest corresponding rope elements from the current rope configuration and final goal rope configuration, and performing consecutive pokes of a fixed small length on the rope element in the direction of the goal configuration element. The detailed procedure are as follows:

### A.2.1    MAZE NAVIGATION TASK

We use the path planning RRT (LaValle, 1998) algorithm to generate optimal paths $P = (p_t, p_{t+1}, p_{t+2}, ...p_n)$ from the current state to the goal state. RRT has privileged information about the obstacle position which is provided to the methods through state. Using these expert paths, we generate state-action expert demonstration dataset for the lower level policy.

### A.2.2    PICK AND PLACE TASK

In order to generate expert demonstrations, we can either use a human expert to perform the pick and place task in virtual reality based Mujoco simulation, or hard code a control policy. We hard-coded the expert demonstrations in our setup. In this task, the robot firstly picks up the block using robotic gripper, and then takes it to the target goal position. Using these expert trajectories, we generate expert demonstration dataset for the lower level policy.

### A.2.3    BIN TASK

In order to generate expert demonstrations, we can either use a human expert to perform the bin task in virtual reality based Mujoco simulation, or hard code a control policy. We hard-coded the expert demonstrations in our setup. In this task, the robot firstly picks up the block using robotic gripper, and then places it in the target bin. Using these expert trajectories, we generate expert demonstration dataset for the lower level policy.

### A.2.4 HOLLOW TASK

In order to generate expert demonstrations, we can either use a human expert to perform the hollow task in virtual reality based Mujoco simulation, or hard code a control policy. We hard-coded the expert demonstrations in our setup. In this task, the robotic gripper has to pick up the square hollow block and place it such that a vertical structure on the table goes through the hollow block. Using these expert trajectories, we generate expert demonstration dataset for the lower level policy.

### A.2.5 ROPE MANIPULATION ENVIRONMENT

We hand coded an expert policy to automatically generate expert demonstrations $e = (s_0^e, s_1^e, \ldots, s_{T-1}^e)$, where $s_i^e$ are demonstration states. The states $s_i^e$ here are rope configuration vectors. The expert policy is explained below.

Let the starting and goal rope configurations be $sc$ and $gc$. We find the cylinder position pair $(sc_m, gc_m)$ where $m \in [1, n]$, such that $sc_m$ and $gc_m$ are farthest from each other among all other cylinder pairs. Then, we perform a poke $(x, y, \theta)$ to drag $sc_m$ towards $gc_m$. The $(x, y)$ position of the poke is kept close to $sc_m$, and poke direction $\theta$ is the direction from $sc_m$ towards $gc_m$. After the poke execution, the next pair of farthest cylinder pair is again selected and another poke is executed. This is repeatedly done for $k$ pokes, until either the rope configuration $sc$ comes within $\delta$ distance of goal $gc$, or we reach maximum episode horizon $T$. Although, this policy is not the perfect policy for goal based rope manipulation, but it still is a good expert policy for collecting demonstrations $\mathcal{D}$. Moreover, as our method requires states and not primitive actions (pokes), we can use these demonstrations $\mathcal{D}$ to collect good higher level subgoal dataset $\mathcal{D}_g$ using primitive parsing.

### A.3 ENVIRONMENT AND IMPLEMENTATION DETAILS

Here, we provide extensive environment and implementation details for various environments. We perform the experiments on two system each with Intel Core i7 processors, equipped with 48GB RAM and Nvidia GeForce GTX 1080 GPUs. We use 28 expert demos for franks kitchen task and 100 demos in all other tasks, and provide the procedures for collecting expert demos for all tasks in Appendix A.2. We empirically increased the number of demonstrations until there was no significant improvement in the performance. In our experiments, we use Soft Actor Critic (Haarnoja et al., 2018b). The actor, critic and discriminator networks are formulated as 3 layer fully connected networks with 512 neurons in each layer.

When calculating $p$, we normalize $Q_{\pi^L}$ values of a trajectory before comparing with $Q_{thresh}$: $((Q_{\pi^L}(s_0^e, s_i^e, a_i) - min\_value)/max\_value) * 100$ for $i = 1$ to $T - 1$. The experiments are run for $4.73e5$, $1.1e5$, $1.32E5$, $1.8E5$, $1.58e6$, and $5.32e5$ timesteps in maze, pick and place, bin, hollow, rope and kitchen respectively. The regularization weight hyper-parameter $\Psi$ is set at $0.001$, $0.005$, $0.005$, $0.005$, $0.005$, and $0.005$, the population hyper-parameter $p$ is set to be $1.1e4$, $2500$, $2500$, $2500$, $3.9e5$, and $1.4e4$, and distance threshold hyper-parameter $Q_{thresh}$ is set at $10$, $0$, $0$, $0$, $0$, and $0$ for maze, pick and place, bin, hollow, rope and kitchen tasks respectively.

In maze navigation, a 7-DOF robotic arm navigates across randomly generated four room mazes, where the closed gripper (fixed at table height) has to navigate across the maze to the goal position. In pick and place task, the 7-DOF robotic arm gripper has to navigate to the square block, pick it up and bring it to the goal position. In bin task, the 7-DOF robotic arm gripper has to pick the square block and place the block inside the bin. In hollow task, the 7-DOF robotic arm gripper has to pick a square hollow block and place it such that a fixed vertical structure on the table goes through the hollow block. In rope manipulation task, a deformable soft rope is kept on the table and the 7-DoF robotic arm performs pokes to nudge the rope towards the desired goal rope configuration. The rope manipulation task involves learning challenging dynamics and goes beyond prior work on navigation-like tasks where the goal space is limited.

In the kitchen task, the 9-DoF franka robot has to perform a complex multi-stage task in order to achieve the final goal. Although many such permutations can be chosen, we formulate the following task: the robot has to first open the microwave door, then switch on the specific gas knob where the kettle is placed. In maze navigation, upper level predicts a subgoal, and the lower level primitive travels in a straight line towards the predicted goal. In pick and place, bin and hollow tasks, we design three primitives, gripper-reach: where the gripper goes to given position $(x_i, y_i, z_i)$, gripper-

open: opens the gripper, and gripper-close: closes the gripper. In kitchen environment, we use the action primitives implemented in RAPS (Dalal et al., 2021). While using RAPS baseline, we hand designed action primitives, which we provide in detail in Section A.3.

### A.3.1 Maze navigation task

In this environment, a 7-DOF robotic arm gripper navigates across random four room mazes. The gripper arm is kept closed and the positions of walls and gates are randomly generated. The table is discretized into a rectangular $W * H$ grid, and the vertical and horizontal wall positions $W_P$ and $H_P$ are randomly picked from $(1, W - 2)$ and $(1, H - 2)$ respectively. In the four room environment thus constructed, the four gate positions are randomly picked from $(1, W_P - 1)$, $(W_P + 1, W - 2)$, $(1, H_P - 1)$ and $(H_P + 1, H - 2)$. The height of gripper is kept fixed at table height, and it has to navigate across the maze to the goal position(shown as red sphere).

The following implementation details refer to both the higher and lower level polices, unless otherwise explicitly stated. The state and action spaces in the environment are continuous. The state is represented as the vector $[p, \mathcal{M}]$, where $p$ is current gripper position and $\mathcal{M}$ is the sparse maze array. The higher level policy input is thus a concatenated vector $[p, \mathcal{M}, g]$, where $g$ is the target goal position, whereas the lower level policy input is concatenated vector $[p, \mathcal{M}, s_g]$, where $s_g$ is the sub-goal provided by the higher level policy. The current position of the gripper is the current achieved goal. The sparse maze array $\mathcal{M}$ is a discrete $2D$ one-hot vector array, where 1 represents presence of a wall block, and 0 absence.

In our experiments, the size of $p$ and $\mathcal{M}$ are kept to be 3 and 110 respectively. The upper level predicts subgoal $s_g$, hence the higher level policy action space dimension is the same as the dimension of goal space of lower primitive. The lower primitive action $a$ which is directly executed on the environment, is a 4 dimensional vector with every dimension $a_i \in [0, 1]$. The first 3 dimensions provide offsets to be scaled and added to gripper position for moving it to the intended position. The last dimension provides gripper control(0 implies a fully closed gripper, 0.5 implies a half closed gripper and 1 implies a fully open gripper). We select 100 randomly generated mazes each for training, testing and validation. For selecting train, test and validation mazes, we first randomly generate 300 distinct mazes, and then randomly divide them into 100 train, test and validation mazes each. We use off-policy Soft Actor Critic (Haarnoja et al., 2018b) algorithm for optimizing RL objective in our experiments.

### A.3.2 Pick and place, Bin and Hollow Environments

In the pick and place environment, a 7-DOF robotic arm gripper has to pick a square block and bring/place it to a goal position. We set the goal position slightly higher than table height. In this complex task, the gripper has to navigate to the block, close the gripper to hold the block, and then bring the block to the desired goal position. In the bin environment, the 7-DOF robotic arm gripper has to pick a square block and place it inside a fixed bin. In the hollow environment, the 7-DOF robotic arm gripper has to pick a hollow plate from the table and place it on the table such that its hollow center goes through a fixed vertical pole placed on the table.

In all the three environments, the state is represented as the vector $[p, o, q, e]$, where $p$ is current gripper position, $o$ is the position of the block object placed on the table, $q$ is the relative position of the block with respect to the gripper, and $e$ consists of linear and angular velocities of the gripper and the block object. The higher level policy input is thus a concatenated vector $[p, o, q, e, g]$, where $g$ is the target goal position. The lower level policy input is concatenated vector $[p, o, q, e, s_g]$, where $s_g$ is the sub-goal provided by the higher level policy. The current position of the block object is the current achieved goal.

In our experiments, the sizes of $p$, $o$, $q$, $e$ are kept to be 3, 3, 3 and 11 respectively. The upper level predicts subgoal $s_g$, hence the higher level policy action space and goal space have the same dimension. The lower primitive action $a$ is a 4 dimensional vector with every dimension $a_i \in [0, 1]$. The first 3 dimensions provide gripper position offsets, and the last dimension provides gripper control (0 means closed gripper and 1 means open gripper). While training, the position of block object and goal are randomly generated (block is always initialized on the table, and goal is always above the table at a fixed height). We select 100 random each for training, testing and validation. For selecting train, test and validation mazes, we first randomly generate 300 distinct environments

with different block and target goal positions, and then randomly divide them into 100 train, test and validation mazes each. We use off-policy Soft Actor Critic (Haarnoja et al., 2018b) algorithm for the `RL` objective in our experiments.

### A.3.3 ROPE MANIPULATION ENVIRONMENT

In the robotic rope manipulation task, a deformable rope is kept on the table and the robotic arm performs pokes to nudge the rope towards the desired goal rope configuration. The task horizon is fixed at 25 pokes. The deformable rope is formed from 15 constituent cylinders joined together. The following implementation details refer to both the higher and lower level polices, unless otherwise explicitly stated. The state and action spaces in the environment are continuous. The state space for the rope manipulation environment is a vector formed by concatenation of the intermediate joint positions. The upper level predicts subgoal $s_g$ for the lower primitive. The action space of the poke is $(x, y, \eta)$, where $(x, y)$ is the initial position of the poke, and $\eta$ is the angle describing the direction of the poke. We fix the poke length to be $0.08$.

While training our hierarchical approach, we select 100 randomly generated initial and final rope configurations each for training, testing and validation. For selecting train, test and validation configurations, we first randomly generate 300 distinct configurations, and then randomly divide them into 100 train, test and validation mazes each. We use off-policy Soft Actor Critic (Haarnoja et al., 2018b) algorithm for optimizing `RL` objective in our experiments.

### A.3.4 IMPACT STATEMENT

Our proposed approach and algorithm are not anticipated to result in immediate technological advancements. Instead, our main contributions are conceptual, targeting fundamental aspects of Hierarchical Reinforcement Learning (`HRL`). By introducing primitive-enabled regularization, we present a novel framework that we believe holds significant potential to advance `HRL` research and its related fields. This conceptual groundwork lays the foundation for future investigations and could drive progress in `HRL` and associated domains.

### A.4 ABLATION EXPERIMENTS

Here, we present the ablation experiments in all six task environments. The ablation analysis includes comparison with `HAC-demos` and `HBC` (Hierarchical behavior cloning) (Figure 14), choosing `RPL` $Q_{thresh}$ hyperparameter (Figure 8), $p$ hyperparameter (Figure 9), `RPL` window size $k$ hyperparameter (Figure 10), learning weight hyperparameter $\phi$ (Figure 11), comparisons with varying number of expert demonstrations used during relabeling and training (Figure 12), comparison with `HER-BC` ablation (Figure 13), effect of sub-optimal demonstrations (Figure 15).

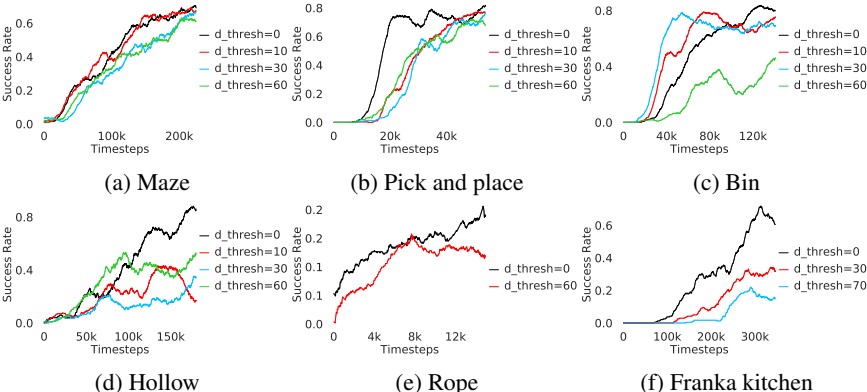

Figure 8: The success rate plots show the performance of PEAR for various values of $Q_{thresh}$ parameter versus number of training timesteps.

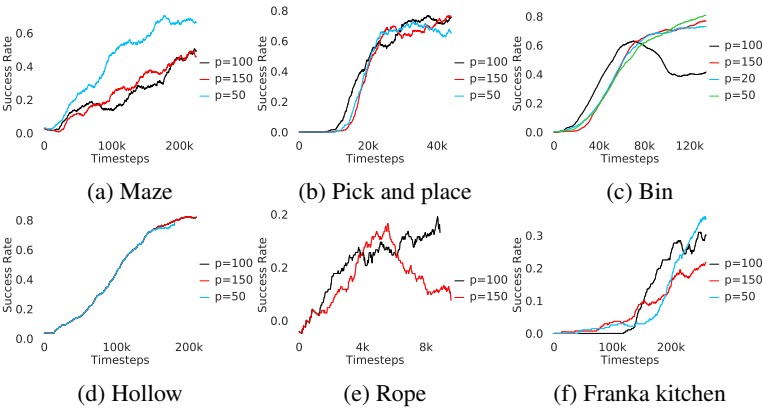

Figure 9: The success rate plots show the performance of PEAR for various values of population number $p$ parameter versus number of training timesteps.

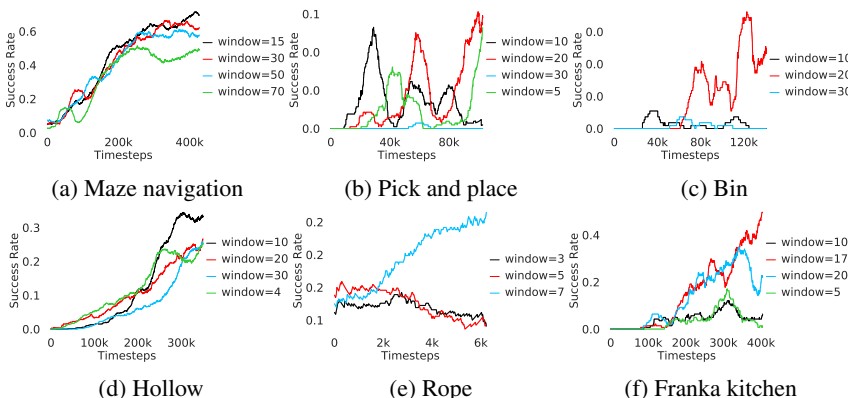

Figure 10: The success rate plots show the performance of RPL for values of $k$ window size parameter versus number of training epochs.

## A.5 QUALITATIVE VISUALIZATIONS

In this subsection, we provide visualizations for various environments.

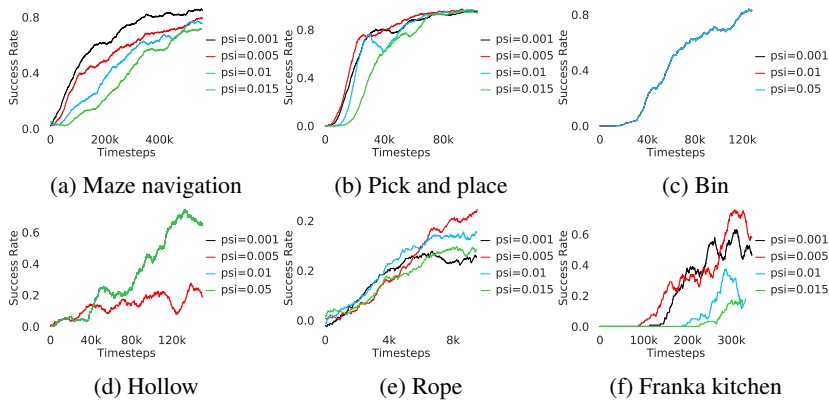

Figure 11: The success rate plots show performance of PEAR for values of learning weight parameter $\psi$ versus number of training timesteps.

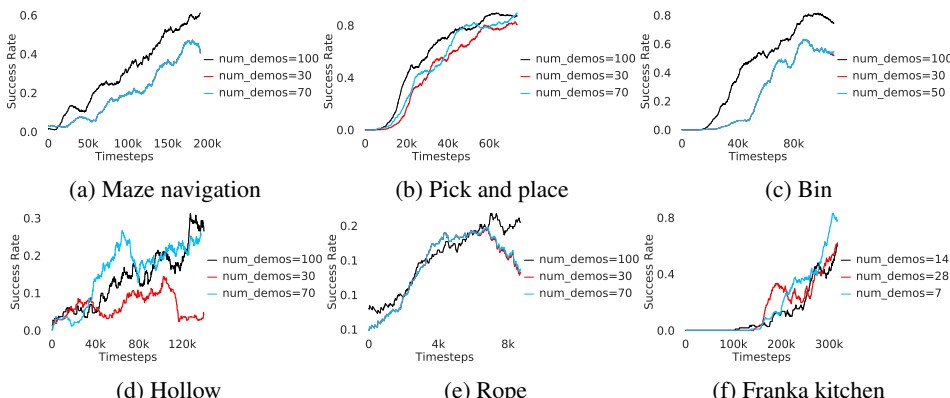

Figure 12: The success rate plots show success rate performance plots of varying number of expert demonstrations versus number of training epochs.

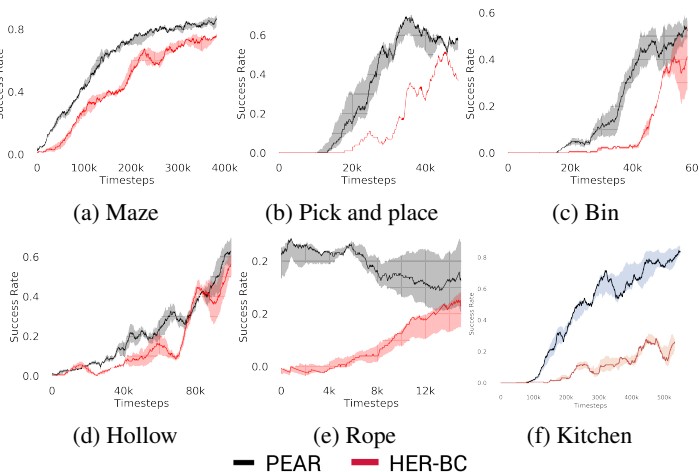

Figure 13: **Comparison with HER-BC baseline**: The figure depicts success rates plots of PEAR-IRL compared with HER-BC baseline, which is a single-level implementation of Hindsight Experience Replay (HER) with expert demonstrations. As can be seen, PEAR consistently outperforms this baseline, which clearly demonstrates the advantages of using our hierarchical formulation in such complex tasks.

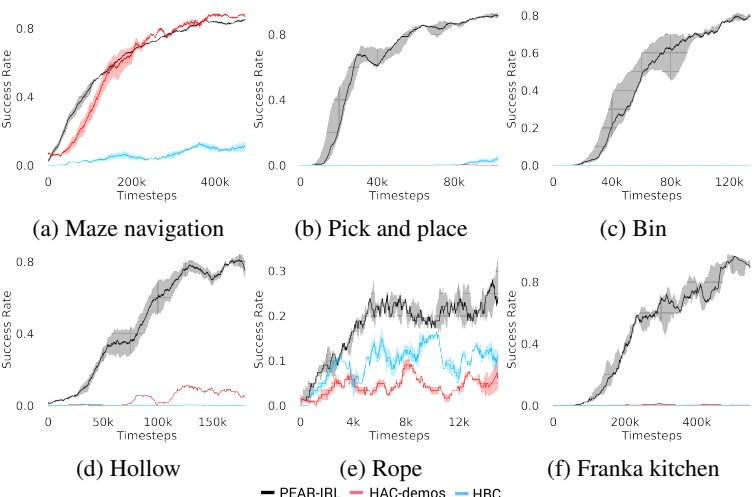

Figure 14: **Success rate comparison: `PEAR-IRL` vs `HAC-demos` vs `HBC`** This figure compares the success rate performances of `PEAR-IRL` with `HAC-demos`, and `HBC` on six sparse maze navigation and manipulation tasks. `HAC-demos` uses hierarchical actor critic (Levy et al., 2018) as the RL objective and is jointly optimized using additional behavior cloning objective, where the lower level uses primitive expert demonstrations and the upper level uses subgoal demonstrations extracted using fixed window based approach (as in RPL (Gupta et al., 2019)). `HBC` (Hierarchical behavior cloning) uses the same demonstrations as `HAC-demos` at both levels, but it is trained using only behavior cloning objective (thus, it does not employ RL). As seen in figure, although `HAC-demos` shows good performance in the easier maze navigation environment, both `HAC-demos` and `HBC` fail to solve the tasks in harder environments, and `PEAR-IRL` significantly outperforms both the baselines. The solid line and shaded region represent the mean and range of success rates across 5 seeds.

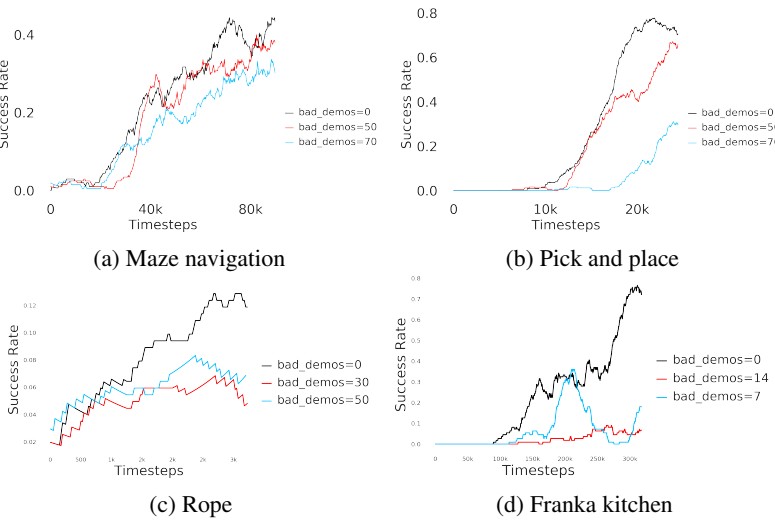

Figure 15: **Ablation with sub-optimal demonstrations:** The success rate plots show the performance of PEAR-IRL with varying number of sub-optimal demonstrations in the expert demonstration dataset. As can be seen, the performance suffers with increasing number of sub-optimal demonstrations.

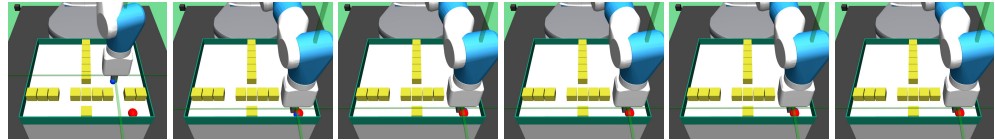

Figure 16: **Successful visualization**: The visualization is a successful attempt at performing maze navigation task

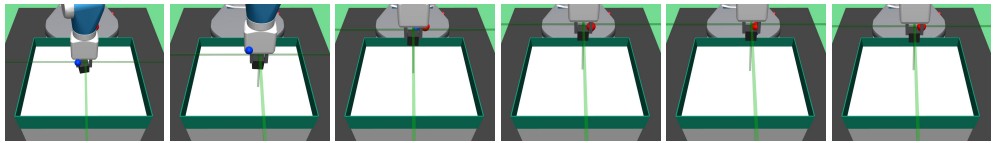

Figure 17: **Successful visualization**: The visualization is a successful attempt at performing pick navigation task

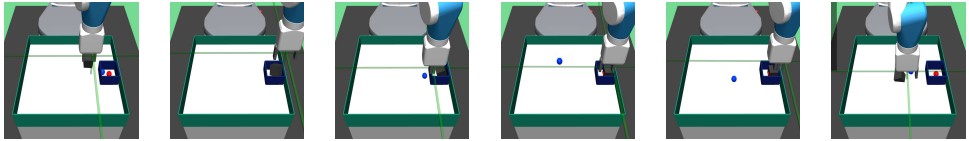

Figure 18: **Successful visualization**: The visualization is a successful attempt at performing bin task

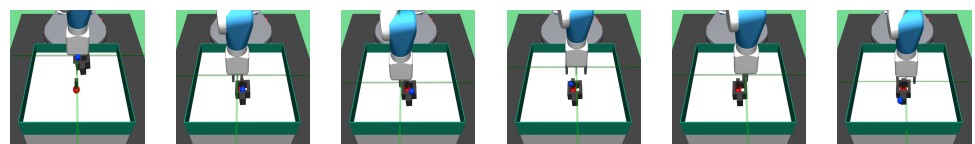

Figure 19: **Successful visualization**: The visualization is a successful attempt at performing hollow task

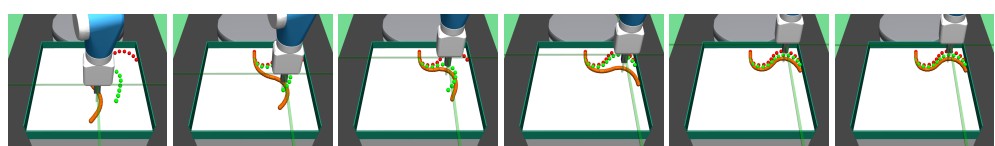

Figure 20: **Successful visualization**: The visualization is a successful attempt at performing rope navigation task

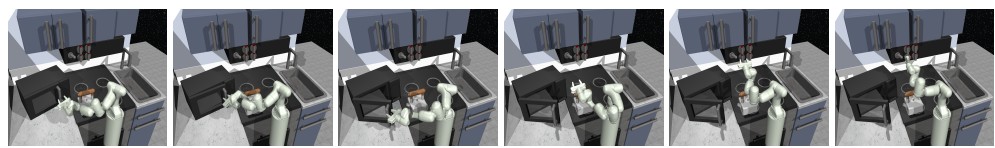

Figure 21: **Successful visualization**: The visualization is a successful attempt at performing kitchen navigation task

