# OpenReview forum: "PEAR: Primitive Enabled Adaptive Relabeling for Boosting Hierarchical Reinforcement Learning"
_ICLR.cc/2025/Conference — ICLR 2025 Poster_

### Official Review · Reviewer_GCp5 · 2024-10-22

**Soundness:** 3
**Presentation:** 2
**Contribution:** 2
**Rating:** 5
**Confidence:** 4

**Summary:**

This paper proposes a method to improve HRL, utilizing a few expert demonstrations. The key insight is that subgoals selected from the dataset can effectively guide exploration for the policy. An adaptive relabeling method is proposed to select the proper subsequent subgoal based on the Q value of the low-level policy. The relabeled data provides an imitation-based regularization for the high-level policy, encouraging it to output reachable, high-quality subgoals for the low-level policy. Experiments in diverse simulation robotic tasks demonstrate the effectiveness of the method.

**Strengths:**

1. Using a few expert demonstrations to improve goal-based HRL is promising. The proposed adaptive relabeling method for IL regularization is straightforward, well-motivated, and yields good results.
2. The paper includes some theoretical analysis.
3. The experiments cover a variety of robotic tasks, including real-world test.

**Weaknesses:**

1. Inconsistent definition: Section 3 states that the expert data contains only states. But in Section 4.2, the low-level regularization term uses actions from the expert data.
2. Addressing the non-stationarity issue in HRL is a main claim of the paper. However, the proposed method does not resolve this issue. The high-level policy still faces non-stationarity, as the transitions in its replay buffer, which are determined by the low-level policy, continue to change throughout the training process.

**Questions:**

1. Many existing works [1,2,3] also use IL loss from data to regularize high-level policy learning. What differentiates the proposed regularization method from these approaches?
2. Why can we set the threshold of Q to 0 (Section 4.1) in all experiments? I believe this hyperparameter should vary, depending on the reward function specific to each task.

[1] Pertsch, et al. "Accelerating reinforcement learning with learned skill priors." Conference on robot learning. PMLR, 2021.
[2] Shi, et al. "Skill-based model-based reinforcement learning." arXiv preprint arXiv:2207.07560 (2022).
[3] Yuan, et al. "Pre-training goal-based models for sample-efficient reinforcement learning." The Twelfth International Conference on Learning Representations. 2024.

---

> ### Author Response · Authors · 2024-11-20
> **Author Response**
>
> We would like to express our gratitude to the reviewer for dedicating their valuable time and effort towards reviewing our manuscript. We deeply appreciate the insightful feedback provided, and we have thoroughly responded to reviewer’s inquiries in the responses provided below.
>
> > **Weakness 1:** Inconsistent definition: Section 3 states that the expert data contains only states. But in Section 4.2, the low-level regularization term uses actions from the expert data.
>
> **Response to Weakness 1:** We agree with the reviewer and apologize for this oversight. We have corrected this in Section 3 of the rebuttal pdf [**rebuttal_pdf_link**](https://openreview.net/pdf?id=0nJEgNpb4l), and assume access to a small number of directed expert demonstrations $D=(e^i)_{i=1}^N$,
>
> where $e^i=(s^e_0,a^e_0, s^e_1,a^e_1 \ldots, s^e_{T-1},a^e_{T-1})$.
>
>
> > **Weakness 2:** Addressing the non-stationarity issue in HRL is a main claim of the paper. However, the proposed method does not resolve this issue. The high-level policy still faces non-stationarity, as the transitions in its replay buffer, which are determined by the low-level policy, continue to change throughout the training process.
>
> **Response to Weakness 2:** We appreciate the reviewer's deep understanding and insight.  We agree that the transitions in its replay buffer still continue to change, and therefore may cause non-stationarity issue. In this work, however, we focus on mitigating non-stationarity by first adaptively relabeling a few expert demonstrations to generate efficient subgoal supervision, and then jointly optimizing HRL agents by employing reinforcement learning (RL) and imitation learning (IL).
>
> We show in Figure 3, 4 and 5 that this indeed mitigates non-stationarity in HRL. However, our **primitive enabled adaptive relabeling** and subsequent **joint optimization** based approach can be added on top of approaches like $\texttt{HAC}$, which also relabel replay buffer transitions to handle non-stationarity. This is an interesting research direction, which we would like to explore in the future.
>
>
> > **Question 1:** Many existing works [1,2,3] also use IL loss from data to regularize high-level policy learning. What differentiates the proposed regularization method from these approaches?
>
> **Response to Question 1:** The two main contributions of the proposed approach are $(i)$ adaptively relabeling expert demonstrations to generate efficient subgoal supervision, and $(ii)$ jointly optimizing using RL and IL using the generated subgoals. Although several prior approaches employ IL regularization, our approach has two major distinctions:
> 1. Our **primitive enabled adaptively relabeling** approach uses the lower level $Q$ function to generate efficient subgoals that are achievable by the lower level policy. These generated subgoals are used to regularize the higher level policy using IL regularization.
> 2. Our approach employs a joint objective (RL and IL). Since it employs RL, $\texttt{PEAR}$ is able to explore the environment for high reward predictions. Further, the IL term regularizes the learnt higher level policy to predict achievable subgoals, thereby mitigating non-stationarity.
>
> > **Question 2:** Why can we set the threshold of Q to 0 (Section 4.1) in all experiments? I believe this hyperparameter should vary, depending on the reward function specific to each task.
>
> **Response to Question 2:** We would like to clarify that we experimentally found the value of $Q_{thresh}=0$ to work well in the tasks, which shows that the training stability is not hyper-volatile with respect to this hyper-parameter. Further, before comparing with $Q_{thresh}$, the $Q_{\pi^{L}}$ values are normalized using the following equation: $(Q_{\pi^{L}}-\text{min value})/(\text{max value} - \text{min value} )*100$, where $\text{min value}$ and $\text{max value}$ are the minimum and maximum values of $Q_{\pi^{L}}$ respectively. Thus, although we do vary $Q_{thresh}$ depending on the reward function specific to each task, we found that out of those values, $Q_{thresh}=0$ works well for all the tasks.
>
> We hope that the responses address the reviewer's concerns. Please let us know, and we will be happy to address additional concerns if any.

---

> ### Author Response · Authors · 2024-11-22
> **A Gentle Reminder**
>
> Dear Reviewer
>
> This is a gentle reminder that we have submitted the rebuttal to address your comments. We sincerely appreciate your feedback and are happy to address any additional questions you may have during this discussion period. We thank you again for taking the time to review our work.
>
> Best regards,
>
> Authors

---

> ### Author Response · Authors · 2024-11-25
> **Follow up request [deadline approaching]**
>
> Dear Reviewer,
>
> We thank you once again for your time and efforts in reviewing our work and providing feedback on your rebuttal. This is a gentle reminder to revise our scores if you find it appropriate, and we are happy to address any additional remaining concerns. We are grateful for your service to the community.
>
> Regards,
>
> Authors

---

> ### Author Response · Authors · 2024-11-28
> **Urgent Request for Reviewer Response**
>
> As per the reviewer's kind feedback and concerns, we have thoroughly provided our responses in the rebuttal and improved the final manuscript. We hope that we have sufficiently addressed the reviewer's concerns.
>
> After our rebuttal, reviewer di4o maintained a score of 8, and reviewers Cpmf and u7xN both increased their scores to 6 for the paper. Since the deadline is approaching, we urgently request the reviewer to respond to our rebuttal and revise our scores if deemed appropriate. We sincerely hope that our work adds sufficient value to the research community, are we are happy to address any additional or remaining concerns. Once again, we are thankful for the time and efforts of the reviewer, which has allowed us to further strengthen the manuscript.

---

> ### Author Response · Authors · 2024-12-01
> **Extremely Urgent Request to the Reviewer**
>
> Dear Reviewer,
>
> We thank you once again for your time and efforts in reviewing our work and providing feedback on your rebuttal. Since the deadline is approaching, we urgently request you to revise our scores if the rebuttal has satisfactorily addressed your concerns. After the rebuttal, reviewer di4o maintained a score of 8, and reviewers Cpmf and u7xN both increased their scores to 6 for the paper.  We will be happy to address any additional remaining concerns.
>
> Regards,
>
> Authors

---

### Official Review · Reviewer_di4o · 2024-11-02

**Soundness:** 3
**Presentation:** 4
**Contribution:** 3
**Rating:** 8
**Confidence:** 4

**Summary:**

PEAR combines its key feature, adaptive goal relabeling, with IL regularization (either MSE or IRL), and other tricks (e.g. margin classification objective) to beat several prior HRL methods on a standard array of tasks using expert demonstrations.

**Strengths:**

The authors carefully outline their construction of the PEAR algorithm, justifying the usage of each component (goal relabeling, the general algorithm, the joint optimization framework, etc) thoroughly and clearly. The sub optimality analysis provides further credence to their method. The method is novel, and notably outperforms previous HRL works (while, in some cases, removing the need for e.g. hand-made action primitives).

I agree with the author view that the significance of the work should be gauged less by its immediate improvement over other LfD methods, and more by its conceptual groundwork. In this regard, this paper is well-written, the findings are well-presented, and the extensive ablations provide further insight into key aspects of the method.

**Weaknesses:**

As authors note, the method is currently reliant on expert demonstrations. However, many benchmarks exist which include other kinds of demonstrations, including human teleop. While the method may not perform well on these demonstrations just yet (as it is listed as an aim of future work), providing results on suboptimal demonstrations would help demonstrate concretely the strong and weak points of authors' method, and potentially provide insights on why it fails in these settings.

Furthermore, it seems inaccurate to state that PEAR uses only “a handful” of demonstrations, when Fig. 13 shows that generally 50-70+ demonstrations are needed to solve the provided tasks (with the exception of Franka Kitchen, which provides fewer demos).

**Questions:**

Have you tested on suboptimal demonstrations? Are there any interesting findings/results which may point to future avenues for research or failings of the method which should be investigated/improved upon?

Were you able to find notable reasons for why the MSE-regularized learning objective would occasionally outperform the IRL-regularized version? Is there any relationship to task difficulty, data diversity, etc?

In “Algorithm 2: PEAR,” line 8, shouldn’t the lower-level policy’s IL regularization be done with $D_g$? Since we are providing state $s^f$ from the goal dataset $D_g$ and subgoal supervision $s^e_g$ to the goal-conditioned low-level policy, then the policy predicts action $a$, and we regularize this to be close to the dataset action $a^f$ (either with MSE or the IRL objective)?

---

> ### Author Response · Authors · 2024-11-20
> **Author Response**
>
> We are thankful to the reviewer for dedicating their valuable time and effort towards evaluating our manuscript. We deeply appreciate the insightful feedback provided, and we have thoroughly responded to reviewer’s inquiries in the responses provided below.
>
> > **Weakness 1:** As authors note, the method is currently reliant on expert demonstrations. However, many benchmarks exist which include other kinds of demonstrations, including human teleop. While the method may not perform well on these demonstrations just yet (as it is listed as an aim of future work), providing results on suboptimal demonstrations would help demonstrate concretely the strong and weak points of authors' method, and potentially provide insights on why it fails in these settings.
>
> **Response to Weakness 1:** We thank the reviewer for this insight, and agree that experiments on suboptimal demonstrations might provide further intuitions on why it might fail in such settings and lead to potential future research problems. To this end, we perform additional experiments with sub-optimal demonstrations and provide them in Figure 16 of the rebuttal pdf [**rebuttal_pdf_link**](https://openreview.net/pdf?id=0nJEgNpb4l). Here are the findings:
> 1. We find that with the increasing number of sub-optimal demonstrations (referred as bad demos in the figure), the performance degrades (which is expected since the IL regularization term becomes sub-optimal).
> 2. We see from the results that the performance degradation with sub-optimal demonstrations becomes more prominent as the environments become harder, which implies that the imitation learning regularization term is crucial for mitigating non-stationarity in harder tasks.
> 3. Since our approach employs joint optimization objectives (**RL** and **IL** objectives), the **IL** objective might generate sub-optimal results when sub-optimal demonstrations are used. Therefore, we would like to explore how to adaptively set the regularization weight parameter in future work, one that is able to adjust (minimize) the **IL** objective weight in the presence of sub-optimal demonstrations.
>
>
> > **Weakness 2:** Furthermore, it seems inaccurate to state that PEAR uses only “a handful” of demonstrations, when Fig. 13 shows that generally 50-70+ demonstrations are needed to solve the provided tasks (with the exception of Franka Kitchen, which provides fewer demos).
>
> **Response to Weakness 2:** We realise now that using the word “handful” may lead to confusion, and we have removed the keyword from the rebuttal pdf [**rebuttal_pdf_link**](https://openreview.net/pdf?id=0nJEgNpb4l).
>
>
> > **Question 1:** Have you tested on suboptimal demonstrations? Are there any interesting findings/results which may point to future avenues for research or failings of the method which should be investigated/improved upon?
>
> **Response to Question 1:** Please refer to our response for Weakness 1.
>
>
> > **Question 2:** Were you able to find notable reasons for why the MSE-regularized learning objective would occasionally outperform the IRL-regularized version? Is there any relationship to task difficulty, data diversity, etc?
>
> **Response to Question 2:** We did further experiments on additional seeds (shown in Figure 3 in the rebuttal pdf [**rebuttal_pdf_link**](https://openreview.net/pdf?id=0nJEgNpb4l)) and found that MSE is only able to outperform IRL based regularization in rope manipulation environment. We believe that although IRL regularization consistently yields good results, it is difficult to train in rope environment due to its unique task structure. Moreover, the expert demonstrations in the rope manipulation environment might be sub-optimal, making it more challenging to learn the IRL regularization objective for this task. In future work, we aim to investigate efficient methods to further boost performance on this environment.
>
>
> > **Question 3:** In “Algorithm 2: PEAR,” line 8, shouldn’t the lower-level policy’s IL regularization be done with $D_g$? Since we are providing state $s^f$ from the goal dataset $D_g$ and subgoal supervision $s_g^e$ to the goal-conditioned low-level policy, then the policy predicts action $a$ , and we regularize this to be close to the dataset action $a^f$ (either with MSE or the IRL objective)?
>
> **Response to Question 3:** We thank the reviewer for this comment and agree that this is indeed a typo. The reviewer is right and the lower-level policy’s IL regularization be done with $D^L_g$ (as also mentioned in Eqn 2). We have corrected this typo in the rebuttal pdf [**rebuttal_pdf_link**](https://openreview.net/pdf?id=0nJEgNpb4l).
>
> We hope that the responses address the reviewer's concerns. Please let us know, and we will be happy to address additional concerns if any.

---

> > ### Comment · Reviewer_di4o · 2024-11-22
> >
> > The authors have sufficiently addressed my questions and concerns, hence I maintain my suggestion to accept with a score of 8.

---

> > > ### Author Response · Authors · 2024-11-22
> > > **Author Response**
> > >
> > > We are thankful to the reviewer for the insightful comments, prompt response, and dedicating their valuable time and effort towards evaluating our manuscript, which has allowed us to strengthen the manuscript.

---

### Official Review · Reviewer_Cpmf · 2024-11-03

**Soundness:** 2
**Presentation:** 3
**Contribution:** 2
**Rating:** 6
**Confidence:** 4

**Summary:**

The authors introduce an algorithm, Primitive Enabled Adaptive Relabeling (PEAR), to address the issue of non-stationary transition and reward functions when implementing HRL.  Like Relay Policy Learning (RPL) (Gupta 2019), PEAR encourages the high level policy to output feasible subgoals using imitation learning.  The main difference from RPL is that instead of imitating the actions from a dataset that occur at fixed intervals, PEAR uses a heuristic to select the latest subgoal that the low level policy can achieve.  The authors show in a variety of experiments that PEAR can outperform several baselines.

Gupta et al. Relay Policy Learning. 2019

**Strengths:**

- The writing was clear and easy to understand.
- The authors included several ablation experiments that provided some important insights.
- The paper included some real world robotic experiments.

**Weaknesses:**

My main concern with this approach is that the contribution seems to be marginal as I do not see a compelling reason why PEAR would consistently perform better than (a) HAC (Levy 2019) with replay buffer augmented with demonstration data and (b) hierarchical behavior cloning, in which pure imitation learning is applied to both a high and low level policies.

HAC already addresses the problem of nonstationarity through relabeling and subgoal testing.  The problem with HAC is that it does not have a built-in mechanism to explore but this can be remedied with the demonstration data that is provided to PEAR.  An advantage of HAC + demonstration data over PEAR, which would use a pure RL objective, is that if the demonstration data is suboptimal, it would not have an imitation learning regularization term forcing the agent to output suboptimal subgoals.  HAC has also demonstrated it can learn more than two levels of hierarchy.  The results of HER+BC, which I understood to be HER with a replay buffer augmented with demonstration data, was often ultimately able to match the performance of PEAR (see Figure 14), making it more likely that HAC, which is close to a hierarchical version of HER, should be able to match PEAR.

In addition, it seems that a pure hierarchical imitation learning approach, in which both levels are trained with supervised learning should also work, at least potentially with more data.  The baseline BC may not have worked well because the tasks were too long horizon, but the addition of a high level policy trained with imitation learning should help.

Levy et al. Hierarchical Actor-Crtiic. 2017

**Questions:**

1.  Why is the Q_{Thresh} set to 0?  If the low-level reward is -1 for not achieving the goal and 0 for achieving the goal, the Q value should be negative for any correct action that puts the agent on a path to achieve the goal.
2.  Is the high level policy only trained with a dataset containing expert trajectories?  Or is it also trained on its own interaction data?  If it is trained on its own interaction data, is there any relabeling done on that?
3.  I did not understand the “margin” component of the objective.  Can you provide another explanation of this component?
4.  Why does the paper characterize the number of demonstrations as a “handful” of demonstrations, when it uses 100 for most tasks?
5.  Have you experimented on any domains with image observations?

I am willing to raise my score if the authors can (i) provide some principled reasons why PEAR should outperform HAC+demonstrations and hierarchical behavior cloning and (ii) provide some answers to the above questions.

---

> ### Author Response · Authors · 2024-11-20
> **Author Response Part 1/2**
>
> We are thankful to the reviewer for dedicating their valuable time and effort towards evaluating our manuscript. We deeply appreciate the insightful feedback provided, and we have thoroughly responded to reviewer’s inquiries in the responses provided below.
>
> > **Weakness 1:** My main concern with this approach is that the contribution seems to be marginal as I do not see a compelling reason why PEAR would consistently perform better than (a) HAC (Levy 2019) with replay buffer augmented with demonstration data
>
> **Response to Weakness 1:** We thank the reviewer for raising this concern. Before stating our response, we would respectfully like to clarify the following:
> 1. $\texttt{PEAR}$ uses our novel **primitive enabled adaptive relabeling** approach to leverage expert demonstrations and generate efficient subgoal supervision for the higher level policy.
> 2. $\texttt{PEAR}$ uses a **joint optimization** objective that employs reinforcement learning (**RL**) and additional imitation learning (**IL**) regularization. RL allows the hierarchical policies to continuously explore the environment for high reward predictions, and IL regularizes the higher level objective to predict achievable subgoals for the lower primitive (and regularizes the lower level primitive to predict efficient primitive actions).
>
> **$\texttt{HAC}$ Limitation:** Although $\texttt{HAC}$ tries to deal with non-stationarity using relabeling and subgoal testing as mentioned by the reviewer, it fails to mitigate non-stationarity in complex long horizon tasks (as seen in Figure 3).
>
> **$\texttt{PEAR}$ can explore using RL:** Further, although $\texttt{PEAR}$ employs imitation learning regularization, it also uses RL in our joint objective formulation which allows it to explore the environment for high reward predictions, even in presence of sub-optimal demonstrations.
>
> **Comparison with HER-BC:** We would also like to point out that although $\texttt{HER-BC}$ baseline works well in easier tasks, it is unable to outperform $\texttt{PEAR}$ and perform well in harder long horizon tasks. Further, we would like to point out that $\texttt{HAC}$ additionally suffers from non-stationarity issue in HRL, since it is hierarchical (unlike the non-hierarchical $\texttt{HER-BC}$).
>
>
> **Additional experiments:** To support our claims, we perform additional experiments: $(i)$ HAC with demos ($\texttt{HAC-demos}$) and $(ii)$ Hierarchical behavior cloning ($\texttt{HBC}$) in **Figure 15** in the rebuttal pdf [**rebuttal_pdf_link**](https://openreview.net/pdf?id=0nJEgNpb4l) to demonstrate the efficacy of $\texttt{PEAR}$ over these baselines.
>
> We would like to point out that it is not straight-forward to train the higher level in hierarchical approaches with behavior cloning, since we typically do not have access to higher level subgoal supervision ($\texttt{PEAR}$ uses primitive enabled adaptive relabeling to acquire efficient subgoal supervision). Therefore, for the following experiments, we employ subgoals extracted using fixed window based approach (as in Relay Policy Learning approach $\texttt{RPL}$):
>
> 1. $\texttt{HAC-demos}$ uses hierarchical actor critic with the RL objective and is jointly optimized using additional behavior cloning objective, where the lower level uses primitive expert demonstrations and the upper level uses subgoal demonstrations extracted using fixed window based approach (as in $\texttt{RPL}$).
> 2. $\texttt{HBC}$ (Hierarchical behavior cloning) uses the same demonstrations as $\texttt{HAC-demos}$ at both levels, but is trained using only behavior cloning objective (thus, does not employ RL).
>
> As seen in **Figure 15**, although $\texttt{HAC-demos}$ shows good performance in the easier maze navigation environment, both $\texttt{HAC-demos}$ and $\texttt{HBC}$ fail to solve the tasks in harder environments, and $\texttt{PEAR-IRL}$ significantly outperforms both the baselines, demonstrating the efficacy of our **primitive enabled adaptive relabeling** and **joint optimization** based approach. Note that both $\texttt{HAC-demos}$ and $\texttt{HBC}$ suffer from non-stationarity issue in HRL, which accounts for their poor performance. This shows that our approach is able to efficiently mitigate non-stationarity in HRL. Finally, **primitive enabled adaptive relabeling** and subsequent **joint optimization** can be added on top of $\texttt{HAC}$, which is an interesting direction we would like to explore in future work.

---

> ### Author Response · Authors · 2024-11-20
> **Author Response Part 2/2**
>
> > **Question 1:** Why is the Q_{Thresh} set to 0? If the low-level reward is -1 for not achieving the goal and 0 for achieving the goal, the Q value should be negative for any correct action that puts the agent on a path to achieve the goal.
>
> **Response to Question 1:** We would like to apologize for the confusion and clarify that before comparing with $Q_{thresh}$, the $Q_{\pi^{L}}$ values are normalized using the following equation: $(Q_{\pi^{L}}-\text{min value})/(\text{max value} - \text{min value} )*100$, where $\text{min value}$ and $\text{max value}$ are the minimum and maximum values of $Q_{\pi^{L}}$ respectively. Since the $Q_{\pi^{L}}$ values are normalized before comparison, $Q_{thresh}=0$ is feasible. We would like to clarify that we experimentally found the value of $Q_{thresh}=0$ to work well in the tasks, which shows that the training stability is not hyper-volatile with respect to this hyper-parameter.
>
>
> > **Question 2:** Is the high level policy only trained with a dataset containing expert trajectories? Or is it also trained on its own interaction data? If it is trained on its own interaction data, is there any relabeling done on that?
>
> **Response to Question 2:** The higher level policy is trained using joint optimization, where it employs reinforcement learning (**RL**) using its own interaction data (enabling efficient exploration), and additional imitation learning (**IL**) regularization. In this work, we do not perform relabeling on interaction data, but we would like to explore this direction in future work.
>
>
> > **Question 3:** I did not understand the “margin” component of the objective. Can you provide another explanation of this component?
>
> **Response to Question 3:** In our **primitive enabled adaptive relabeling** procedure, we employ the lower level $Q$ function $Q_{\pi^{L}}$ to select efficient subgoals, which is conditioned on the expert states $s^e$. Since $Q_{\pi^{L}}$ is the lower level $Q$ function and is simultaneously trained, it might over-estimate and provide erroneous values for states $s^e_{unseen}$ that are unseen during previous training (a known issue with Q functions in RL). Therefore, we posit that the over-estimation issue can be avoided by adding a margin objective that penalizes the $Q$ values on out-of-distribution states.  We state this as margin classification objective in the paper and empirically found this objective to stabilize training.
>
>
> > **Question 4:** Why does the paper characterize the number of demonstrations as a “handful” of demonstrations, when it uses 100 for most tasks?
>
> **Response to Question 4:** We realize now that using the word “handful” may lead to confusion, and we have removed the keyword from the rebuttal pdf [**rebuttal_pdf_link**](https://openreview.net/pdf?id=0nJEgNpb4l).
>
>
> > **Question 5:** Have you experimented on any domains with image observations?
>
> **Response to Question 5:** We did experiment with image observations in the maze navigation, pick and place, and rope environments, and found that $\texttt{PEAR}$ consistently shows impressive performance on the tasks and outperforms the baselines. However, due to resource and time constraints, we decided to not add those results to this paper draft. We will try to finish the experiments in time, and add them to the final paper draft submission.
>
> We hope that the responses address the reviewer's concerns. Please let us know, and we will be happy to address additional concerns if any.

---

> ### Author Response · Authors · 2024-11-22
> **A Gentle Reminder**
>
> Dear Reviewer
>
> This is a gentle reminder that we have submitted the rebuttal to address your comments. We sincerely appreciate your feedback and are happy to address any additional questions you may have during this discussion period. We thank you again for taking the time to review our work.
>
> Best regards,
>
> Authors

---

> ### Author Response · Authors · 2024-11-25
> **Follow Up Request [deadline approaching]**
>
> Dear Reviewer,
>
> We thank you once again for your time and efforts in reviewing our work and providing feedback on your rebuttal. This is a gentle reminder to revise our scores if you find it appropriate, and we are happy to address any additional remaining concerns. We are grateful for your service to the community.
>
> Regards,
>
> Authors

---

> ### Comment · Reviewer_Cpmf · 2024-11-26
>
> Thank you authors for the response to my questions.  I increased my score by a point as I believe the approach is sufficiently new.  However, I still do not understand why HAC is characterized as nonstationary.  The reward for subgoal actions in HAC never depend on the value functions of states achieved by the current low level policy, they only depend on the value functions of states achieved by fully trained low level policies as a result of the hindsight transitions.  The problem with HAC is not nonstationarity but rather exploration.  Your HAC+demos experiment does not seem to do what I requested, which was a pure HAC with the replay buffers supplemented with the fixed window transitions from the demonstrations rather than incorporating the imitation learning objective objective into HAC.  I also do not understand why a pure hierarchical behavior cloning approach, in which high level policy is trained to output states reached from demonstrations, would be "nonstationary".  This objective would have no dependence on the states reached by the current low level policies.

---

> > ### Author Response · Authors · 2024-11-27
> > **Further Author Response**
> >
> > We are thankful to the reviewer for increasing the score and for the insightful feedback. We agree with the reviewer and apologize for erroneously mentioning that HAC-demos and HBC still suffer from non-stationarity. However, these methods still lead to poor performance due to the issue of unachievable subgoals prediction. We explain this in detail below:
> >
> > > **Question 1**: I still do not understand why HAC is characterized as nonstationary. The reward for subgoal actions in HAC never depend on the value functions of states achieved by the current low level policy, they only depend on the value functions of states achieved by fully trained low level policies as a result of the hindsight transitions. The problem with HAC is not nonstationarity but rather exploration.
> >
> > **Response to Question 1**: HAC addresses the problem of non-stationarity through *replay buffer relabeling* and *subgoal testing*. However, HAC's *replay buffer relabeling* assumes the achieved subgoal to be the predicted subgoal, and accordingly re-evaluates sparse rewards. Thus, HAC may learn to predict subgoal $g_t$ that is not achievable by the current lower level policy, leading to poor performance. PEAR is able to mitigate this by employing **primitive enabled adaptive relabeling** to always predict achievable subgoals for the lower level primitive. We also demonstrate this in our extensive experiments on complex and sparsely-rewarded long horizon tasks, where PEAR is able to significantly outperform HAC.
> >
> >
> > > **Question 2**: Your HAC+demos experiment does not seem to do what I requested, which was a pure HAC with the replay buffers supplemented with the fixed window transitions from the demonstrations rather than incorporating the imitation learning objective objective into HAC.
> >
> > **Response to Question 2**: We would also like to clarify that we did implement pure HAC approach with the replay buffer appended with fixed window transitions from the expert demonstrations, like the reviewer had requested. However, in order to ensure fair comparisons with our approach, we also incorporated the imitation learning objective (IL) with HAC objective. We found this IL objective to improve the performance when compared to pure HAC based approach.
> >
> >
> > > **Question 3**: I also do not understand why a pure hierarchical behavior cloning approach, in which high level policy is trained to output states reached from demonstrations, would be "nonstationary". This objective would have no dependence on the states reached by the current low level policies.
> >
> > **Response to Question 3**: In hierarchical BC approach, the higher level is trained using fixed window based subgoals from expert demonstrations. Notably, the fixed window approach may select **unachievable subgoals** for the lower level policy. Due to this, the lower level policy might not be able to achieve the predicted subgoal $g_t$. Further, since HBC only uses imitation learning to train hierarchical policies, it is unable to efficiently explore the environment, leading to sub-optimal predictions. These reasons account for the poor performance of HBC baseline, as demonstrated by our experiments. In contrast, our **adaptive relabeling** based approach always selects achievable subgoals according to the lower level policy, and is able to efficiently explore the environment using RL.
> >
> >
> > We hope that we have been able to clarify the reviewer's concerns, and if deemed appropriate, we request the reviewer to kindly increase our score. We are grateful to the reviewer for the service to the community.

---

> > ### Author Response · Authors · 2024-12-01
> > **Follow up request [deadline approaching]**
> >
> > Dear Reviewer,
> >
> > We thank you once again for your time and efforts in reviewing our work and providing feedback on your rebuttal. This is a gentle reminder to let us know if we have satisfactorily addressed the reviewer's concerns, and to revise our scores if you find it appropriate. We are happy to address any additional remaining concerns. We are grateful for your service to the community.
> >
> > Regards,
> >
> > Authors

---

### Official Review · Reviewer_u7xN · 2024-11-04

**Soundness:** 3
**Presentation:** 2
**Contribution:** 2
**Rating:** 6
**Confidence:** 3

**Summary:**

The paper presents a novel approach called Primitive Enabled Adaptive Relabeling (PEAR) aimed at enhancing Hierarchical Reinforcement Learning (HRL) for complex long-horizon tasks. The authors propose a two-phase methodology where the first phase involves adaptive relabeling of expert demonstrations to generate subgoals, followed by joint optimization of HRL agents through Reinforcement Learning (RL) and Imitation Learning (IL). The results indicate that PEAR outperforms various baselines in both synthetic and real-world robotic tasks, achieving up to 80% success rates in challenging environments.

**Strengths:**

1. **Innovative Approach:** The use of adaptive relabeling to generate subgoals tailored to the capabilities of the lower primitive is a significant contribution that addresses the non-stationarity issue in HRL.
2. **Theoretical Justification:** The authors provide theoretical analysis that bounds the sub-optimality of their approach, lending credibility to their claims.
3. **Comprehensive Experiments:** Extensive experiments across multiple challenging tasks demonstrate the practical efficacy of PEAR, showing improved performance and sample efficiency over existing methods.
4. **Real-World Application:** The validation of PEAR in real-world tasks enhances the relevance and applicability of the research.

**Weaknesses:**

1. **Expert Demonstrations Requirement:** The first phase of PEAR relies on expert demonstrations to generate subgoals, which raises concerns about the fairness of comparison with other HRL methods that do not require such demonstrations. This could affect the generalizability of the findings.
2. **Lack of Recent Comparisons:** The paper does not include comparisons with several hierarchical reinforcement learning methods published in the last three years. This omission limits the contextual relevance of the results and could misrepresent the state of the art, such as [1], [2].

[1] Kim J, Seo Y, Shin J. Landmark-guided subgoal generation in hierarchical reinforcement learning[J]. Advances in neural information processing systems, 2021, 34: 28336-28349.

[2] Wang, Vivienne Huiling, et al. "State-conditioned adversarial subgoal generation." Proceedings of the AAAI conference on artificial intelligence. Vol. 37. No. 8. 2023.

3. **Citation Issues:** Many citations are sourced from CoRR instead of their formal conference versions. This oversight should be corrected to ensure academic integrity and accurate referencing. For example, the last two references should be cited as follows (Note that there are many other citation errors beyond these two):

Wulfmeier, Markus, et al. "Data-efficient hindsight off-policy option learning." International Conference on Machine Learning. PMLR, 2021.

Zhang, Tianren, et al. "Generating adjacency-constrained subgoals in hierarchical reinforcement learning." Advances in Neural Information Processing Systems 33 (2020): 21579-21590.

**Questions:**

- How do the authors justify the reliance on expert demonstrations in the first phase of PEAR compared to HRL methods that function without such requirements?
- Can the authors provide additional comparisons with recent HRL approaches to strengthen the positioning of PEAR within the current landscape?

---

> ### Author Response · Authors · 2024-11-20
> **Author Response Part 1/2**
>
> We would like to express our gratitude to the reviewer for dedicating their valuable time and effort towards evaluating our manuscript, which has allowed us to strengthen the manuscript. We deeply appreciate the insightful feedback provided, and we have thoroughly responded to reviewer's inquiries in the responses provided below.
>
> > **Weakness 1:** Expert Demonstrations Requirement: The first phase of PEAR relies on expert demonstrations to generate subgoals, which raises concerns about the fairness of comparison with other HRL methods that do not require such demonstrations. This could affect the generalizability of the findings.
>
> **Response to Weakness 1:** We agree with the reviewer that the proposed approach relies on *expert demonstrations* to generate subgoals, which can be challenging in environments where generating such *demonstrations* is computationally expensive (we also mention this in the discussion section of the paper).
>
> However, we would like to respectfully point out that:
> * **collecting a few *expert demonstrations* is often feasible** in such robotics control tasks, and indeed several prior works employ such *expert demonstrations* (covered in detail in the Related Work section).
> * the main contribution of this work is to use **primitive enabled adaptive relabeling** to generate efficient subgoals, and subsequently leveraging them in our **joint optimization** based approach.
>
> Although we compare our approach with several baselines that employ expert demonstrations (Relay Policy Learning $\texttt{RPL}$, Discriminator Actor Critic $\texttt{DAC}$, Behavior Cloning $\texttt{BC}$), the purpose of comparisons with baselines that do not use expert demonstrations are as follows:
> * $\texttt{HAC}$: to show that $\texttt{PEAR}$ is able to better mitigate non-stationarity.
> * $\texttt{RAPS}$: to show that $\texttt{PEAR}$ outperforms approaches that use behavior priors.
> * $\texttt{HIER}$ and $\texttt{HIER-NEG}$: to show the importance of extracting subgoals using **primitive enabled adaptive relabeling** and subsequent **joint optimization**.
> * $\texttt{FLAT}$: to show the importance of our hierarchical formulation (temporal abstraction and improved exploration).
>
> Thus, we would like to respectfully point out that such comparisons offer additional insights into into the factors contributing to the superior performance of $\texttt{PEAR}$ compared to the baselines.
>
>
> **Additional Experiments:** Further, in response to reviewer Cpmf, we perform additional experiments comparing our approach to two more baselines that employ expert demonstrations  (Figure 15 in the rebuttal pdf [**rebuttal_pdf_link**](https://openreview.net/pdf?id=0nJEgNpb4l)).
>
> 1. $\texttt{HAC-demos}$ uses hierarchical actor critic with the RL objective and is jointly optimized using additional behavior cloning objective, where the lower level uses primitive expert demonstrations and the upper level uses subgoal demonstrations extracted using fixed window based approach (as in $\texttt{RPL}$).
> 2. $\texttt{HBC}$ (Hierarchical behavior cloning) uses the same demonstrations as $\texttt{HAC-demos}$ at both levels, but is trained using only behavior cloning objective (thus, does not employ RL).
>
> As seen in Figure 15, $\texttt{PEAR-IRL}$ significantly outperforms both the baselines, demonstrating the efficacy of our **primitive enabled adaptive relabeling** and **joint optimization** based approach.
>
>
> > **Weakness 2:** Lack of Recent Comparisons: The paper does not include comparisons with several hierarchical reinforcement learning methods published in the last three years. This omission limits the contextual relevance of the results and could misrepresent the state of the art, such as [1], [2].
>
> **Response to Weakness 2:** Based on the reviewer's concern, we have performed and added additional experiments in the paper to compare our approach with $\texttt{SAGA}$ [1]. $\texttt{SAGA}$ employs state conditioned discriminator network training to address non-stationarity, by ensuring that the high-level policy generates subgoals that align with the current state of the low-level policy. We provide the comparisons in Figure 3 of the rebuttal pdf [**rebuttal_pdf_link**](https://openreview.net/pdf?id=0nJEgNpb4l).
>
> We find that although $\texttt{SAGA}$ performs well in maze navigation environment, it fails to solve harder tasks, where $\texttt{PEAR}$ is able to significantly outperform the baseline. This demonstrates that $\texttt{SAGA}$ suffers from non-stationarity issue in harder long horizon tasks, and $\texttt{PEAR}$ is able to better mitigate non-stationarity issue and effectively solve long horizon tasks.
>
>
> > **Weakness 3:** Citation Issues: Many citations are sourced from CoRR ... and accurate referencing.
>
> **Response to Weakness 3:** We thank the reviewer for pointing this out. We have corrected all the citations with their formal venue citations in the rebuttal pdf [**rebuttal_pdf_link**](https://openreview.net/pdf?id=0nJEgNpb4l).

---

> ### Author Response · Authors · 2024-11-20
> **Author Response Part 2/2**
>
> > **Question 1:** How do the authors justify the reliance on expert demonstrations in the first phase of PEAR compared to HRL methods that function without such requirements?
>
> **Response to Question 1:** Please refer to our response to Weakness 1.
>
>
> > **Question 2:** Can the authors provide additional comparisons with recent HRL approaches to strengthen the positioning of PEAR within the current landscape?
>
> **Response to Question 2:** We have performed additional experiments and provided in the rebuttal pdf [**rebuttal_pdf_link**](https://openreview.net/pdf?id=0nJEgNpb4l). Please refer to our response to Weakness 2.
>
> [1] Wang, Vivienne Huiling, et al. "State-conditioned adversarial subgoal generation." Proceedings of the AAAI conference on artificial intelligence. Vol. 37. No. 8. 2023.
>
> We hope that the responses address the reviewer's concerns. Please let us know, and we will be happy to address additional concerns if any.

---

> > ### Comment · Reviewer_u7xN · 2024-11-23
> >
> > Thank you for addressing my concerns. I appreciate the improvements made, and I have raised the score accordingly. However, there are still two points that I believe need further attention:
> >
> > 1. Figure 1 needs to be redrawn. The text is too small, and it does not clearly convey the innovations of the work. I suggest revisiting the figure to make it more informative and legible.
> >
> > 2. Citations remain an issue. I encourage you to approach the references with greater rigor. Many citations are from ArXiv, but there are official conference versions available, such as:
> >
> > Vezhnevets, Alexander Sasha, et al. "Feudal networks for hierarchical reinforcement learning." In: International Conference on Machine Learning, PMLR, 2017, pp. 3540-3549.
> >
> > Zhang, Tianren, et al. "Generating adjacency-constrained subgoals in hierarchical reinforcement learning." Advances in Neural Information Processing Systems, 2020, 33: 21579-21590.

---

> > > ### Author Response · Authors · 2024-11-24
> > > **Author Response**
> > >
> > > We are thankful to the reviewer for the insightful comments, prompt response, and dedicating their valuable time and effort towards evaluating our manuscript, which has allowed us to strengthen the manuscript. We are also glad that we could address the reviewer's concerns. Further,
> > > 1. We have re-drawn Figure 1 to improve the text size and to make it more legible.
> > > 2. We apologize for failing to fix the citations earlier, and we have fixed the citations according to their formal venues.
> > >
> > > We hope that the responses address the reviewer's concerns. Please let us know, and we will be happy to address additional concerns if any.

---

> > > ### Author Response · Authors · 2024-12-01
> > > **Follow Up Request [deadline approaching]**
> > >
> > > Dear Reviewer,
> > >
> > > We thank you once again for your time and efforts in reviewing our work and providing feedback on your rebuttal. This is a gentle reminder to let us know if we have satisfactorily addressed the reviewer's concerns, and to revise our scores if you find it appropriate. We are happy to address any additional remaining concerns. We are grateful for your service to the community.
> > >
> > > Regards,
> > >
> > > Authors

---

> ### Author Response · Authors · 2024-11-22
> **A Gentle Reminder**
>
> Dear Reviewer
>
> This is a gentle reminder that we have submitted the rebuttal to address your comments. We sincerely appreciate your feedback and are happy to address any additional questions you may have during this discussion period. We thank you again for taking the time to review our work.
>
> Best regards,
>
> Authors

---

### Author Response · Authors · 2024-12-03
**General Response to Area Chairs and Reviewers**

We thank all the reviewers for their valuable feedback and insightful comments. We are particularly encouraged that the reviewers consider our proposed approach **novel and promising** (u7xN, di4o and GCp5), our **theoretical analysis insightful** (u7xN and GCp5), our **empirical evaluation and ablations impressive and extensive** (u7xN, Cpmf, GCp5, and di4o), the **paper well-written and well-organized** (Cpmf, di4o), and our **real world experiments credible** (u7xN, Cpmf, GCp5). We address individual questions of reviewers in separate responses. We have uploaded a modified version of our paper based on reviewers' suggestions: [**rebuttal_pdf_link**](https://openreview.net/pdf?id=0nJEgNpb4l).

We first enlist additional experiments to address the reviewer's concerns for your consideration.

### Additional Experiments.

* **Comparison with state-of-the-art baseline**: In order to address reviewer **u7xN**'s concerns, we implemented recent state-of-the-art approach **SAGA** (State-conditioned adversarial subgoal generation) and critically compared it with PEAR. We demonstrate that PEAR is able to significantly outperform this approach, which shows that PEAR efficiently mitigates non-stationarity in HRL and demonstrates impressive performance over state-of-the-art approaches. We are happy that reviewer acknowledged the efficacy of this experiment, and kindly raised the score to 6.
* **Additional Hierarchical baselines: HAC-demos and HBC**:  In order to address reviewer **Cpmf**'s concerns, we implemented two additional baselines: **HAC-demos** (Hindsight actor critic with expert demonstrations) and **HBC** (Hierarchical behavior cloning) and compared it with PEAR. This comparison shows that *primitive enabled adaptive relabeling* and *joint optimization* are crucial for mitigating non-stationarity in HRL and improved performance. We are happy that reviewer acknowledged the efficacy of this experiment, and kindly raised the score to 6.

We have added the additional experiments and improvements suggested by the reviewers in the submitted manuscript, and provided detailed clarifications to address all the reviewers' concerns in the rebuttal. We also provide a summary of our core contributions below.

### Summary of core contributions:

* In this work, we propose a novel approach PEAR that efficiently mitigates non-stationarity in off-policy HRL for solving complex long horizon tasks.
* Our adaptive relabeling based approach generates efficient higher level subgoal supervision according to the current goal achieving capability of the lower primitive.
* We perform detailed theoretical analysis to bound the sub-optimality of our approach, and to theoretically justify the benefits of periodic re-population using adaptive relabeling.
* We perform extensive experimentation on six sparse robotic environments to empirically demonstrate our superior performance and sample efficiency over prior baselines.
* We show that PEAR demonstrates impressive performance in multiple challenging real world tasks.
* In summary, we propose a novel hierarchical reinforcement learning algorithm that is a promising step towards building practical robotic systems for real-world scenarios.

We greatly appreciate all reviewers' time and effort in reviewing our paper. We hope that our responses and updates to the paper have addressed all the concerns, and solidified **PEAR** as a promising HRL approach for building practical systems to solve complex tasks.

---

### Meta-Review · Area_Chair_U3vn · 2024-12-18

**Metareview:**

This paper proposed a two-stage HRL method that uses expert demonstrations to partition trajectories into subsets of training data for low-level policies. The main claimed contributions are that the method generates efficient subgoal supervision, the method mitigates non-stationarity in HRL, experiments show the work outperforms prior work, and that real-world experiments also demonstrates "impressive performance" in real world tasks. Experiments provide a reasonable amount of evidence of these claims (except the latter, in my opinion, as these experiments were fairly limited). Reviewers were almost all positive about this work, and I found no glaring issues with the paper that were either unmentioned by the reviewers or unsatisfactorily addressed by the authors.

**Additional Comments On Reviewer Discussion:**

Most reviewers meaningfully engaged with the author's responses. However, the reviewer rating below a 6 (Reviewer GCp5) did not. After reading their concerns and the author response to them (notational issues, addressing nonstationarity), the authors' response to these concerns were satisfactory (e.g., the evidence in Fig. 4 in particular supports the claim of mitigating nonstationarity).

---

### Decision · Program_Chairs · 2025-01-22

Accept (Poster)